# Revisiting Convergence: Shuffling Complexity Beyond Lipschitz Smoothness

Qi He [1]  Peiran Yu [2]  Ziyi Chen [1]  Heng Huang [1]

## Abstract

Shuffling-type gradient methods are favored in practice for their simplicity and rapid empirical performance. Despite extensive development of convergence guarantees under various assumptions in recent years, most require the Lipschitz smoothness condition, which is often not met in common machine learning models. We highlight this issue with specific counterexamples. To address this gap, we revisit the convergence rates of shuffling-type gradient methods without assuming Lipschitz smoothness. Using our stepsize strategy, the shuffling-type gradient algorithm not only converges under weaker assumptions but also match the current best-known convergence rates, thereby broadening its applicability. We prove the convergence rates for nonconvex, strongly convex, and non-strongly convex cases, each under both random reshuffling and arbitrary shuffling schemes, under a general bounded variance condition. Numerical experiments further validate the performance of our shuffling-type gradient algorithm, underscoring its practical efficacy.

## 1. Introduction

Gradient-based optimization has always been a critical area due to its extensive practical applications in machine learning, including reinforcement learning (Sutton and Barto, 2018), hyperparameter optimization (Feurer and Hutter, 2019), and large language models (Radford et al., 2018). While numerous gradient-based algorithms have been developed for convex functions (Nemirovskij and Yudin, 1983; Nesterov, 2013; d'Aspremont et al., 2021), research on nonconvex functions has become particularly active in recent years, driven by advances in deep learning. Notably, with unbiased stochastic gradients and bounded variance, SGD

achieves an optimal complexity of $\mathcal{O}(\epsilon^{-4})$ (Ghadimi and Lan, 2013), which matches the lower bound established by Arjevani et al. (2023).

In practice, however, random shuffling-type methods have demonstrated advantages over traditional SGD. These methods, which involve iterating over data in a permuted order rather than drawing i.i.d. samples, are widely used in deep learning frameworks due to their simplicity and computational efficiency. Empirical studies have shown that such methods often outperform standard SGD in terms of convergence speed and generalization ability (Bottou, 2009; 2012). Notably, shuffling-based training has been found to mitigate gradient variance, leading to smoother optimization trajectories compared to purely stochastic updates (Gürbüzbalaban et al., 2021; HaoChen and Sra, 2019).

Theoretical analyses of shuffling-type methods have gained significant attention in recent years, aiming to explain their empirical success while tackling the unique challenges posed by the lack of independence between consecutive updates. Many early works focused on convex optimization, often relying on strong convexity assumptions to establish linear or sublinear convergence rates (Gürbüzbalaban et al., 2021; HaoChen and Sra, 2019; Safran and Shamir, 2020). More recently, research efforts have extended beyond the convex setting (Mishchenko et al., 2020; Nguyen et al., 2021; Koloskova et al., 2023), suggesting that random reshuffling can exhibit faster convergence than SGD even in nonconvex settings under certain conditions, such as when gradient noise is structured or exhibits low variance.

Despite these advances, most theoretical analyses rely on the Lipschitz smoothness assumption, which imposes restrictive upper and lower bounds on the gradient variations. While this assumption holds in many standard optimization settings, it fails to capture a broad class of important machine learning models, including deep language models (Zhang et al., 2019), phase retrieval (Chen et al., 2023), and distributionally robust optimization (Chen et al., 2023). As a result, many practical scenarios remain outside the scope of existing theoretical guarantees. To address this gap, our work develops new techniques to analyze the convergence of shuffling-type gradient algorithms under relaxed smoothness assumptions, aiming to provide a more comprehensive theoretical foundation applicable to a wider range

[1]Department of Computer Science, University of Maryland, College Park. [2]Department of Computer Science and Engineering, University of Texas Arlington. Correspondence to: Qi He <qhe123@umd.edu>, Heng Huang <heng@umd.edu>.

*Proceedings of the $42^{nd}$ International Conference on Machine Learning*, Vancouver, Canada. PMLR 267, 2025. Copyright 2025 by the author(s).

of machine learning models.

We consider the following finite sum minimization problem:

$$\min_{w \in \mathbb{R}^d} \left\{ F(w) := \frac{1}{n} \sum_{i=1}^{n} f(w; i) \right\}, \qquad \text{(P)}$$

where $f(\cdot; i) : \mathbb{R}^d \to \mathbb{R}$ is smooth and possibly nonconvex for $i \in [n] := \{1, \ldots, n\}$. Problem (P) covers empirical loss minimization as a special case, therefore can be viewed as formulation for many machine learning models, such as logistic regression, reinforcement learning, and neural networks.

We summarize our main contributions as follows:

- We proved the convergence of the shuffling-type gradient algorithm under non-uniform smoothness assumptions, where the Hessian norm is bounded by a subquadratic function $\ell$ of the gradient norm. With specific stepsizes and a general bounded variance condition, we achieved a total complexity of $\mathcal{O}(n^{\frac{p+1}{2}} \epsilon^{-3})$ gradient evaluations for the nonconvex case with random reshuffling, and $\mathcal{O}(n^{\frac{p}{2}+1} \epsilon^{-3})$ for arbitrary scheme, where $0 \le p < 2$ is the degree of function $\ell$ . These results match those with Lipschitz smoothness assumptions in Nguyen et al. (2021) when $p = 0$ and $\ell$-smoothness degenerates to Lipschitz smoothness.

- For the strongly convex case, we established a complexity of $\widetilde{\mathcal{O}}(n^{\frac{p+1}{2}} \epsilon^{-\frac{1}{2}})$ for random reshuffling. In the non-strongly convex case, the complexity is $\mathcal{O}(n^{\frac{p+1}{2}} \epsilon^{-\frac{3}{2}})$ for random reshuffling. Without assuming bounded variance, we established complexity of $\widetilde{\mathcal{O}}(n\epsilon^{-\frac{1}{2}})$ for arbitrary scheme in strongly convex case, and $\mathcal{O}(n\epsilon^{-\frac{3}{2}})$ in non-strongly convex case.

- We conducted numerical experiments to demonstrate that the shuffling-type gradient algorithm converges faster than SGD on two important non-Lipschitz smooth applications.

## 2. Preliminaries

### 2.1. Shuffling-Type Gradient Algorithm

In practice, the random shuffling method has demonstrated its superiority over SGD, as shown in Bottou (2009) and Bottou (2012). Specifically, Bottou (2009) shows that shuffling-type methods achieve a convergence rate of approximately $\mathcal{O}(1/T^2)$, where $T$ is the iteration count. Beyond shuffling-type stochastic gradient methods, variants such as SVRG have been applied in various scenarios, including decentralized optimization, as discussed in Shamir (2016) and De and Goldstein (2016).

The analysis of shuffling-type methods has a long history. For convex cases, Gürbüzbalaban et al. (2021) demonstrated that when the objective function is a sum of quadratics or smooth functions with a Lipschitz Hessian, and with a diminishing stepsize, the average of the last update in each epoch of RGA converges strictly faster than SGD with probability one. Additionally, they showed that when the number of epochs $T$ is sufficiently large, the Reshuffling Gradient Algorithm (RGA) asymptotically converges at a rate of $\mathcal{O}(1/T^2)$. Similarly, Nguyen et al. (2021) established a convergence rate of $\mathcal{O}(1/T^2)$ for strongly convex and globally $L$-smooth functions. Furthermore, with uniform sampling and a bounded variance assumption or convexity on each component function, they showed that the convergence rate can be improved to $\mathcal{O}(1/nT^2)$.

In contrast, there is not much research on nonconvex cases. For example, Nguyen et al. (2021) demonstrated a convergence rate of $\mathcal{O}(T^{-2/3})$; Koloskova et al. (2023) proved a convergence rate of $\mathcal{O}\left( \frac{1}{T} + \min\left\{ \left(\frac{n\sigma}{T}\right)^{\frac{2}{3}}, \left(\frac{n\sigma^2}{T}\right)^{\frac{1}{2}} \right\} \right)$ for single shuffling gradient method; Wang et al. (2024) analyzed Adam algorithm with random reshuffling scheme under $(L_0, L_1)$-smoothness but they did not explicitly show the convergence rate.

### 2.2. Counterexamples

Although $L$-smoothness is empirically false in LSTM (Zhang et al., 2019) and Transformers (Crawshaw et al., 2022), we give some concrete counterexamples to demonstrate the popularity of non-Lipschitz functions in this section. First we give two machine learning examples, then we mention some common non-Lipschitz functions.

**Example 1.** The first example is distributionally robust optimization (DRO), which is a popular optimization framework for training robust models. DRO is introduced to deal with the distribution shift between training and test datasets. In (Levy et al., 2020), it is formulated equivalently as follows.

$$\min_{w \in \mathcal{W}, \theta \in \mathbb{R}} L(w, \theta) := \mathbb{E}_{\xi \sim P} \psi^* \left( \frac{\ell(w; \xi) - \theta}{\lambda} \right) + \theta, \quad (1)$$

where $w$ and $\theta$ are the parameters to be optimized, $\xi$ is a sample randomly drawn from data distribution $P$, $\ell(w; \xi)$ is the loss function, $\psi^*$ is the conjugate function of the divergence function $\psi$ we choose to measure the difference between distributions, and $\lambda > 0$ is the regularization coefficient. It is proved in Jin et al. (2021) that $L(w, \theta)$ is not always Lipschitz-smooth even if $\ell(w; \xi)$ is Lipschitz-smooth and the variance is bounded.

**Example 2.** The second example is the phase retrieval problem. Phase retrieval is a nonconvex problem in X-

ray crystallography and diffraction imaging (Drenth, 2007; Miao et al., 1999). The goal is to recover the structure of a molecular object from intensity measurements. Let $x \in \mathbb{R}^d$ be the true object and $y_r = |a_r^\top x|^2$ for $r = 1, \ldots, m$, where $a_r \in \mathbb{R}^d$. The problem is to solve:

$$\min_{z \in \mathbb{R}^d} f(z) := \frac{1}{2m} \sum_{r=1}^{m} (y_r - |a_r^\top z|^2)^2. \qquad (2)$$

This objective function is a high-order polynomial in high-dimensional space, thus it does not belong to the $L$-smooth function class.

**Example 3.** There are many common functions that are not Lipschitz smooth, including polynomial functions with order $> 2$, exponential functions, logarithmic functions and rational functions.

## 2.3. Relaxation of Lipschitz Smoothness

Because of the existence of these counterexamples, people have recently been investigating about smoothness assumptions that are more general than the traditional Lipschitz smoothness. In Zhang et al. (2019), $(L_0, L_1)$-smoothness was proposed as the first relaxed smoothness notion motivated by language modeling. It is defined as below:

**Definition 2.1.** ($(L_0, L_1)$-smoothness) A real-valued differentiable function $f$ is $(L_0, L_1)$-smooth if there exist constants $L_0, L_1 > 0$ such that

$$\|\nabla^2 f(w)\| \leq L_0 + L_1 \|\nabla f(w)\|.$$

Lipschitz smoothness can be viewed as a special case of $(L_0, L_1)$ smoothness when $L_1 = 0$. Under $(L_0, L_1)$-smoothness assumption, various convergence algorithms have been developed including clipped or normalized GD/SGD (Zhang et al., 2019), momentum accelerated clipped GD/SGD (Zhang et al., 2020), ADAM (Wang et al., 2022) and variance-reduced clipping (Reisizadeh et al., 2023) with optimal sample complexity on stochastic non-convex optimization.

Other relaxed smoothness assumptions include asymmetric generalized smoothness motivated by distributionally robust optimization (Jin et al., 2021) and its extension to $\alpha$-symmetric generalized smoothness (Chen et al., 2023) and $\ell$-smoothness (Li et al., 2023a). In this paper, we use the definition of $\ell$-smoothness as below:

**Definition 2.2.** ($\ell$-smoothness) A real-valued differentiable function $f$ is $\ell$-smooth if there exists some non-decreasing continuous function $\ell : [0, +\infty) \to (0, +\infty)$ such that for any $w \in \text{dom}(f)$ and constant $C > 0$,

$\mathcal{B}(w, \frac{C}{\ell(\|\nabla f(w)\| + C)}) \subseteq \text{dom}(f)$; and for any $w_1, w_2 \in \mathcal{B}(w, \frac{C}{\ell(\|\nabla f(w)\| + C)})$,

$$\|\nabla f(w_1) - \nabla f(w_2)\| \leq \ell(\|\nabla f(w)\| + C) \cdot \|w_1 - w_2\|.$$

Another equivalent definition of $\ell$-smoothness in Li et al. (2023a) is:

**Definition 2.3.** A real-valued differentiable function $f$ is $\ell$-smooth for some non-decreasing continuous function $\ell : [0, +\infty) \to (0, +\infty)$ if

$$\|\nabla^2 f(x)\| \leq \ell(\|\nabla f(x)\|)$$

almost everywhere.

In the proof of sections we focus on Definition 2.2, though Definition 2.3 provides a clearer perspective on the relationship between $\ell$-smoothness and other smoothness notions. Both $(L_0, L_1)$-smoothness and traditional Lipschitz smoothness can be seen as special cases of $\ell$-smoothness. It is straightforward to verify that the loss functions in phase retrieval and distributionally robust optimization (DRO) satisfy $\ell$-smoothness. While extensive research has been conducted based on $(L_0, L_1)$-smoothness, comparatively less work has been done under the more general $\ell$-smooth framework. Xian et al. established convergence results for generalized GDA/SGDA and GDmax/SGDmax in minimax optimization, while Zhang et al. (2024) analyzed MGDA for multi-objective optimization under $\ell$-smoothness.

## 3. Algorithm

As demonstrated in our counterexamples, the Lipschitz smoothness assumption does not always hold in problem (P). In such non-Lipschitz scenarios, gradients can change drastically, posing a significant challenge for these algorithms. To address this issue, we propose a new stepsize strategy, detailed in Algorithm 1 and section 4, to improve performance under these challenging conditions. This strategy aims to choose the stepsize to accommodate the variance and instability in gradients, thereby enhancing the robustness of the optimization process.

In this algorithm, we start with an initial point $\tilde{w}_0$. During each iteration $t \in [T]$, either all the samples are shuffled, or we keep the order of the samples as in the last epoch. This reshuffling introduces variance in the order of samples, which can help mitigate issues related to gradient instability. For each step $j \in [n]$, we use the gradient from a single sample with number $\pi_j^{(t)}$ to update the weights $w$. The notation $\pi_j^{(t)}$ is used to denote the $j$-th element of the permutation $\pi^{(t)}$ for $j \in [n]$. Each outer loop through the data is counted as an epoch, and our convergence analysis focuses on the performance after the completion of each full epoch.

**Algorithm 1** Shuffling-type Gradient Algorithm

**Initialization**: Choose an initial point $\tilde{w}_0 \in dom(F)$.
**for** $t = 1, 2, \cdots, T$ **do**
    Set $w_0^{(t)} := \tilde{w}_{t-1}$;
    Generate permutation $\pi^{(t)}$ of $[n]$.
    Compute non-increasing stepsize $\eta_t$.
    **for** $j = 1, \cdots, n$ **do**
        Update $w_j^{(t)} := w_{j-1}^{(t)} - \frac{\eta_t}{n} \nabla f(w_{j-1}^{(t)}; \pi_j^{(t)})$.
    **end for**
    Set $\tilde{w}_t := w_n^{(t)}$.
**end for**

There are multiple strategies to determine $\pi^{(t)}$:

- If $\pi^{(t)}$ is a fixed permutation of $[n]$, Algorithm 1 functions as an incremental gradient method. This method maintains a consistent order of samples, which can simplify the analysis and implementation.

- If $\pi^{(t)}$ is shuffled only once in the first iteration and then used in every subsequent iteration, Algorithm 1 operates as a shuffle-once algorithm. This strategy introduces randomness at the beginning but maintains a fixed order thereafter, providing a balance between randomness and stability.

- If $\pi^{(t)}$ is regenerated in every single iteration, Algorithm 1 becomes a random reshuffling algorithm. This approach maximizes the randomness in the sample order, potentially offering the most robustness against the erratic behavior of non-Lipschitz gradients by constantly changing the sample order.

Although the random reshuffling scheme is most used in practice, each of these strategies offers distinct advantages and can be selected based on the specific requirements and characteristics of the optimization problem at hand. For this reason, we will give convergence rates for random and arbitrary shuffling scheme.

## 4. Convergence Analysis

### 4.1. Main Results

In this section, we present the main results of our convergence analysis. Our findings indicate that, with proper stepsizes, it is possible to achieve the same convergence rate, up to a logarithm difference, as under the Lipschitz smoothness assumption. First, we introduce the following assumptions regarding problem (P). Assumption 4.1 is a standard assumption, and Assumption 4.2 requires all $F$ and $f(\cdot; i)$ to be $\ell$-smooth.

**Assumption 4.1.** $\mathrm{dom}(F) := \{w \in \mathbb{R}^d : F(w) < +\infty\} \neq \emptyset$ and $F^* := \inf_{w \in \mathbb{R}^d} F(w) > -\infty$.

**Assumption 4.2.** $F$ and $f(\cdot; i)$ are $\ell$-smooth for some sub-quadratic function $\ell$, $\forall i \in [n]$.

Here, we assume all functions share the same $\ell$ function without loss of generality, as we can always choose the pointwise maximum of all their $\ell$ functions. We define $p$ to be the degree of the $\ell$ function such that $p = \sup_{p \geq 0}\{p| \lim_{w \to \infty} \frac{\ell(w)}{w^p} > 0\}$. Since $\ell$ is sub-quadratic, we have $0 \leq p < 2$.

Next, we introduce our assumption about the gradient variances.

**Assumption 4.3.** There exist two constants $\sigma, A \in (0, +\infty)$ such that $\forall i \in [n]$,

$$\frac{1}{n}\sum_{i=1}^n \|\nabla f(w; i) - \nabla F(w)\|^2 \leq A\|\nabla F(w)\|^2 + \sigma^2, \quad (3)$$

$a.s.$, $\forall w \in \mathrm{dom}(F)$.

Since component gradients behave as gradient estimations of the full gradient, this assumption can be viewed as a generalization of the more common assumption $\mathbb{E}[\|\nabla F(w; \xi) - \nabla F(w)\|] \leq \sigma^2$, which is used in most $\ell$-smooth work.

#### 4.1.1. NONCONVEX CASE

Let us denote $\Delta_1 := F(w_0^{(1)}) - F^*$. Under Assumptions 4.1 to 4.3, we have the following result for random shuffling scheme. Proofs can be found in Appendix A.1.1 and A.1.2.

**Theorem 4.4.** *Suppose Assumptions 4.1, 4.2 and 4.3 hold, Let $\{\tilde{w}_t\}_{t=1}^T$ be generated by Algorithm 1 with random reshuffling scheme. For any $0 < \delta < 1$, we denote $H := \frac{4\Delta_1}{\delta}$, $G := \sup\{u \geq 0 | u^2 \leq 2\ell(2u) \cdot H\}$, $G' := \sqrt{2(1 + nA)}G + \sqrt{2n}\sigma$, $L := \ell(2G')$. For any $\epsilon > 0$, choose $\eta_t$ and $T$ such that*

$$\eta_t \leq \frac{1}{2L\sqrt{\frac{A}{n} + 1}}, \quad \sum_{t=1}^T \eta_t^3 \leq \frac{n\Delta_1}{L^2\sigma^2}, \quad T \geq \frac{32\Delta_1}{\eta_T \delta \epsilon^2},$$

*then with probability at least $1 - \delta$, we have $\|\nabla F(w_0^{(t)})\| \leq G$ for every $1 \leq t \leq T$*

$$\frac{1}{T}\sum_{t=1}^T \|\nabla F(w_0^{(t)})\|^2 \leq \epsilon^2.$$

*Remark* 4.5. By choosing $\eta_t = \eta = \mathcal{O}(\sqrt[3]{\frac{n^{1-p}}{T}}) = \mathcal{O}(n^{\frac{1-p}{2}}\epsilon)$, we can achieve a complexity of $T = \mathcal{O}(\frac{n^{\frac{p-1}{2}}}{\epsilon^3})$ outer iterations and $\mathcal{O}(\frac{n^{\frac{p+1}{2}}}{\epsilon^3})$ total number of gradient evaluations, ignoring constants, where $p$ is the order of the $\ell$ function in Definition 2.2. As $p$ goes to 0, $\ell$-smoothness degenerates to the traditional Lipschitz smoothness, and

our total number of gradient evaluations goes to $\mathcal{O}(\frac{\sqrt{n}}{\epsilon^3})$ once again, which matches the complexity in Corollary 1 of Nguyen et al. (2021). If $\epsilon \leq 1/\sqrt{n}$, one possible stepsize is $\eta = \frac{\sqrt{n}\epsilon}{2L\sqrt{\frac{A}{n}+1}}$.

Our result here has polynomial dependency on $\frac{1}{\delta}$, $T = \mathcal{O}(\delta^{-\frac{3}{2} - \frac{p}{2-p}})$. It is important to note that, in our setting, $\delta$ accounts for the probability that Lipschitz smoothness does not hold—a consideration absent in standard Lipschitz smoothness settings. Therefore, the dependency here is not as good as in $L$-smoothness cases. In fact, a polynomial dependency on $\delta$ is typical in papers with similar smoothness assumptions, e.g. in Li et al. (2023a), Li et al. (2023b), Xian et al. and Zhang et al. (2024).

Next we consider arbitrary $\pi^{(t)}$ scheme in Algorithm 1.

**Theorem 4.6.** *Suppose Assumptions 4.1, 4.2 and 4.3 hold. Let $\{\tilde{w}_t\}_{t=1}^T$ be generated by Algorithm 1 with arbitrary scheme. Define $H = 2\Delta_1$, $G := \sup\{u \geq 0 | u^2 \leq 2\ell(2u) \cdot H\}$, $G' := \sqrt{2(1+nA)}G + \sqrt{2n}\sigma$, $L := \ell(2G')$. For any $\epsilon > 0$, choose $\eta_t$ and $T$ such that*

$$\eta_t \leq \frac{1}{L\sqrt{2(3A+2)}}, \ \sum_{t=1}^T \eta_t^3 \leq \frac{2\Delta_1}{3\sigma^2 L^2}, \ T \geq \frac{8\Delta_1}{\eta_T \epsilon^2},$$

*then we have $\|\nabla F(w_0^{(t)})\| \leq G$ for every $1 \leq t \leq T$ and*

$$\frac{1}{T}\sum_{t=1}^T \|\nabla F(w_0^{(t)})\|^2 \leq \epsilon^2.$$

This theorem gives the convergence rate for arbitrary scheme in Algorithm 1. By choosing $\eta_t = \eta = \mathcal{O}\left(\sqrt[3]{\frac{1}{n^p T}}\right) = \mathcal{O}\left(\frac{\epsilon}{n^{\frac{p}{2}}}\right)$, we achieve a complexity of $\mathcal{O}\left(\frac{n^{\frac{p}{2}}}{\epsilon^3}\right)$ outer iterations and $\mathcal{O}\left(\frac{n^{\frac{p}{2}+1}}{\epsilon^3}\right)$ total gradient evaluations, ignoring constants. Without the randomness in $\pi$ in every iteration, the complexity's dependency on $n$ is increased by $\mathcal{O}(\sqrt{n})$. One possible stepsize is $\eta = \frac{\epsilon}{L\sqrt{2(3A+2)}}$.

### 4.1.2. STRONGLY CONVEX CASE

For strongly convex case, we give results for both random reshuffling scheme and arbitrary scheme, with constant learning rate. Proof can be found in Appendix A.2.

**Assumption 4.7.** Function $F$ in (P) is $\mu$-strongly convex on $\text{dom}(F)$.

**Theorem 4.8.** *Suppose Assumptions 4.1, 4.2, 4.3 and 4.7 hold. Let $\{\tilde{w}_t\}_{t=1}^T$ be generated by Algorithm 1 with random reshuffling scheme. For any $0 < \delta < 1$, we denote $H := \max\{\frac{3\sigma^2}{4\mu}\log\frac{4}{\epsilon} + \Delta_1, \frac{4\Delta_1}{\delta}\}$, $G := \sup\{u \geq 0 | u^2 \leq 2\ell(2u) \cdot H\}$, $G' := \sqrt{2(1+nA)}G + \sqrt{2n}\sigma$, $L := \ell(2G')$.*

*For any $\epsilon > 0$, if we choose $\eta_t$ and $T$ such that*

$$\eta_t = \eta = \frac{4\log(\sqrt{n}T)}{\mu T}, T \geq 4\sqrt{\frac{\Delta_1}{n\delta\epsilon}}, \frac{T}{\log(\sqrt{n}T)} \geq$$

$$\frac{4}{\mu}\max\left\{2, L\sqrt{2(3A+2)}, L\sigma\sqrt{\frac{8}{n\mu\delta\epsilon}}, \sqrt[3]{\frac{T\sigma^2 L^2}{n\Delta_1}}\right\},$$

*then for any $0 < \delta < 1$, with probability at least $1 - \delta$ we have*

$$F(w_0^{(T+1)}) - F^* \leq \epsilon.$$

In Theorem 4.8, we can achieve a complexity of $\widetilde{\mathcal{O}}\left(n^{\frac{p-1}{2}}\epsilon^{-\frac{1}{2}}\right)$ outer iterations and $\widetilde{\mathcal{O}}\left(n^{\frac{p+1}{2}}\epsilon^{-\frac{1}{2}}\right)$ total gradient evaluations with $\eta = \widetilde{\mathcal{O}}\left(n^{\frac{1-p}{2}}\epsilon^{\frac{1}{2}}\right)$, ignoring constants. This matches the result in Nguyen et al. (2021) with the same assumptions in the degenerate case of $p = 0$. The dependence on $\delta$ is $T = \mathcal{O}(\delta^{-\frac{1}{2} - \frac{p}{2-p}})$.

It is not hard to follow proof of Theorem 4.6 for arbitrary scheme in strongly convex case and achieve a complexity of $\widetilde{\mathcal{O}}\left(n^{\frac{p}{2}+1}\epsilon^{-\frac{1}{2}}\right)$ total gradient evaluations. Here we give a slightly stronger result where we remove Assumption 4.3 to match the corresponding result in Lipschitz smooth case.

**Theorem 4.9.** *Suppose Assumptions 4.1, 4.2 and 4.7 hold. Let $\{\tilde{w}_t\}_{t=1}^T$ be generated by Algorithm 1 with arbitrary scheme. We denote $S = \{w | F(w) \leq F(w_0^{(1)})\}$, $G' = \max_w\{\|\nabla f(w; i)\| | w \in S, i \in [n]\}$, $L := \ell(2G')$. For any $\epsilon > 0$, choose $\eta_t$ and $T$ such that*

$$\eta_t = \eta = \frac{6\log(T)}{\mu T} \leq \frac{\Delta_1 \mu^2}{9(\mu^2 + L^2)\sigma_*^2},$$

$$T = \widetilde{\mathcal{O}}(\epsilon^{-\frac{1}{2}}) \geq \frac{12L^2 \log(T)}{\mu^2},$$

*where $\sigma_*$ is the standard deviation at $w_*$. Then we have $\|\nabla F(w_0^{(t)})\| \leq G'$ and*

$$F(w_0^{(T+1)}) - F^* \leq \epsilon.$$

In Theorem 4.9, we achieve a complexity of $\widetilde{\mathcal{O}}\left(\epsilon^{-1/2}\right)$ outer iterations and $\widetilde{\mathcal{O}}\left(n\epsilon^{-1/2}\right)$ total gradient evaluations with $\eta = \widetilde{\mathcal{O}}\left(n^{-1}\epsilon^{\frac{1}{2}}\right)$, ignoring constants. It should be noted that the constant $G'$ here is implicitly determined by constant $p$ and can potentially be large. Therefore, the complexity here cannot be directly compared with those in Theorem 4.6 or 4.8.

### 4.1.3. NON-STRONGLY CONVEX CASE

Next we consider the case where only non-strongly convexity are assumed. In the following theorem, we assume the

optimal solution exists and denote one as $w_*$, the standard deviation at $w_*$ as $\sigma_* := \sqrt{\frac{1}{n}\sum_{i=1}^{n}\|\nabla f(w_*;i)\|^2}$ and the average value of $\{w_0^{(t)}\}_{t=1}^{T}$ as $\bar{w}_T = \frac{1}{T}\sum_{t=1}^{T}w_0^{(t)}$. Proof for this section can be found in Appendix A.3.

**Assumption 4.10.** Functions $f(\cdot;i)$ in (P) are convex on $\mathrm{dom}(F)$, for all $i \in [n]$.

**Theorem 4.11.** *Suppose Assumptions 4.1, 4.2, 4.3 and 4.10 hold. Let $\{\tilde{w}_t\}_{t=1}^{T}$ be generated by Algorithm 1 with random reshuffling scheme. For any $0 < \delta < 1$, define $H$, $G$, $G'$, $L$ as in Theorem 4.4. For any $\epsilon > 0$, choose $\eta_t = \eta$ and $T$ such that*

$$\eta \leq \min\left\{\frac{1}{2L\sqrt{\frac{A}{n}+1}}, \sqrt[3]{\frac{n\Delta_1}{T\sigma^2 L^2}}, \sqrt[3]{\frac{3n\|w_0^{(1)}-w_*\|^2}{2LT\sigma_*^2}}\right\},$$

$$T \geq \frac{4\|w_0^{(1)}-w_*\|^2}{\eta\delta\epsilon},$$

*then with probability at least $1-\delta$, we have $\|\nabla F(w_0^{(t)})\| \leq G$ for every $1 \leq t \leq T$ and*

$$F(\bar{w}_T) - F^* \leq \epsilon.$$

By choosing $\eta = \mathcal{O}\left(\sqrt[3]{\frac{n^{1-p}}{T}}\right) = \mathcal{O}\left(n^{\frac{1-p}{2}}\epsilon^{0.5}\right)$, we achieve a complexity of $\mathcal{O}\left(\frac{n^{\frac{p-1}{2}}}{\epsilon^{1.5}}\right)$ outer iterations and $\mathcal{O}\left(\frac{n^{\frac{p+1}{2}}}{\epsilon^{1.5}}\right)$ total number of gradient evaluations, ignoring constants. The dependency on $\delta$ is $T = \mathcal{O}(\delta^{-\frac{3}{2}-\frac{p}{2-p}})$. If $\epsilon \leq 1/n$, one possible stepsize is $\eta = \frac{\sqrt{n\epsilon}}{2L\sqrt{\frac{A}{n}+1}}$.

Similarly, we can follow proof of Theorem 4.6 for arbitrary scheme and achieve a complexity of $\mathcal{O}\left(\frac{n^{\frac{p}{2}+1}}{\epsilon^{1.5}}\right)$ total gradient evaluations. Now we give a result without variance assumption 4.3.

**Theorem 4.12.** *Suppose Assumptions 4.1, 4.2 and 4.10 hold. Let $\{\tilde{w}_t\}_{t=1}^{T}$ be generated by Algorithm 1 arbitrary scheme. Define $S = \{w|F(w) \leq F(w_0^{(1)})\}$, $G' = \max_w\{\|\nabla f(w;i)\| \mid w \in S, i \in [n]\} < \infty$, $L = \ell(2G')$. For any $\epsilon > 0$, choose $\eta_t = \eta$ and $T$ such that*

$$\eta \leq \frac{1}{G'}\sqrt{\frac{3\epsilon}{2L}}, T = \mathcal{O}(\epsilon^{-1.5}) \geq \frac{\|w_0^{(1)}-w_*\|^2}{\eta\epsilon},$$

*then we have $\|\nabla F(w_0^{(t)})\| \leq G$ for every $1 \leq t \leq T$ and*

$$\min_{t\in[T]} F(w_T) - F^* \leq \epsilon.$$

By choosing $\eta = \mathcal{O}(\sqrt[3]{\frac{1}{T}})$, we have the complexity of $\mathcal{O}(\frac{1}{\epsilon^{1.5}})$ outer iterations and $\mathcal{O}(\frac{n}{\epsilon^{1.5}})$ total number of gradient evaluations, ignoring constants. One possible stepsize is $\eta = \frac{\sqrt{3\epsilon}}{G'\sqrt{L}}$.

## 4.2. Proof Sketch and Technical Novelty

Broadly speaking, our approach involves two main goals: first, demonstrating that Lipschitz smoothness is maintained with high probability along the training trajectory $\{\tilde{w}_t\}$, and second, showing that, conditioned on Lipschitz smoothness, the summation of gradient norms is bounded with high probability. Here we slightly abuse the term 'Lipschitz' and 'Lipschitz smoothness' to refer to the property between neighboring steps along the training trajectory.

For the first goal, in Lemma A.4, we prove by induction that when starting an iteration with a bounded gradient, the entire training trajectory during this iteration will have bounded gradients. Consequently, we only need to verify the Lipschitz smoothness condition at the start of each iteration. However, at this point, the two goals become intertwined. We need Lipschitz smoothness to bound the gradient differences, but we also need the gradient norm bounds to establish Lipschitz smoothness. Our solution is to address both issues simultaneously.

Assuming that, before a stopping time $\tau$, Lipschitz smoothness holds, we bound the gradient norm up to that time in Lemma A.5. However, this process is nontrivial. Since we are examining behavior before a stopping time, every expectation is now conditioned on $t < \tau$, rendering all previous estimations for shuffling gradient algorithms inapplicable. This presents a contradiction: we want to condition on $t < \tau$ when applying Lipschitz smoothness, but we do not want this condition when estimating other quantities. In Lemma A.5, we find a method to separately handle these two requirements, allowing us to achieve both goals simultaneously.

## 4.3. Limitations and Future works

Although we have proved upper bounds for the complexity of shuffling gradient algorithms, there are certain limitations in our work that we leave for future research:

- First, as is common with many optimization algorithms, it is challenging to verify that the bounds presented are indeed the lower bounds. Future work could explore improving these results, for instance, by reducing the dependency on $\delta$ to a logarithmic factor, or by proving that the current bounds are, in fact, tight lower bounds.

- Second, although we showed results for arbitrary shuffling schemes, there are better results for single shuffling under Lipschitz smoothness, for example Ahn et al. (2020) proved $\mathcal{O}(\frac{1}{nT^2})$ convergence rate for strongly convex objectives. It is interesting to see whether we can achieve the same convergence rate with $\ell$-smoothness as well.

- The results in Theorem 4.9 and Theorem 4.12 depends on constant $G'$ that can be potentially very large and

hard to verify. In the absence of both Lipschitz smoothness and bounded variance, the behavior of gradients can be hard to track. We hope our results here can be a first step for future work.

- Lastly, shuffling gradient methods have been integrated with variance reduction techniques (Malinovsky et al., 2023). Exploring the performance of these algorithms under relaxed smoothness assumptions is another promising direction for future work.

## 5. Numerical Experiments

We compare reshuffling gradient algorithm (Algorithm 1) with SGD on multiple $\ell$-smooth optimization problems to prove its effectiveness. Experiments are conducted with different shuffling schemes, on convex, strongly convex and nonconvex objective functions, including synthetic functions, phase retrieval, distributionally robust optimization (DRO) and image classification.

### 5.1. Convex and Strongly Convex Settings

We first consider convex functions $f_{i,k}(x) = x_i^4 + kx_i$ of $x \in \mathbb{R}^{50}$ for all $(i,k) \in \mathcal{E} := \{1,2,\ldots,50\} \times \{-10,-9,\ldots,9,10\}$, as well as their sample average $f(x) = \frac{1}{1050}\sum_{(k,i)\in\mathcal{E}} f_{i,k}(x) = \frac{1}{50}\sum_{i=1}^{50} x_i^4$. It can be easily verified that $f$ and all $f_{i,k}$ are convex but not strongly convex, and $\ell$-smooth (with $\ell(u) = 3u^{2/3}$) but not Lipschitz-smooth. Then we compare reshuffling gradient algorithm (Algorithm 1) with SGD on the objective $\min_{x\in\mathbb{R}^{50}} f(x)$. Specifically, for each SGD update $x \leftarrow x - \eta\nabla f_{k,i}(x)$, $(k,i) \in \mathcal{E}$ is obtained uniformly at random. For Algorithm 1, we adopt three shuffling schemes as elaborated in Section 3. The fixed-shuffling scheme and shuffling-once fix all permutations $\pi^{(t)}$ respectively to be the natural sequence $(1,-10),(1,-9),\ldots(50,10)$ and its random permutation at the beginning, while the uniform-shuffling scheme obtains permutations $\pi^{(t)}$ uniformly at random and independently for all iterations $t$. We implement each algorithm 100 times with initialization $x_0 = [1,\ldots,1]$ and fine-tuned stepsizes 0.01 (i.e., $\eta = 0.01$ for SGD and $\frac{\eta_t}{n} = 0.01$ for Algorithm 1), which takes around 3 minutes in total. We plot the learning curves of $f(x_t)$ averaged among the 100 times, as well as the 95% and 5% percentiles in the left of Figure 1, which shows that Algorithm 1 with all shuffling schemes converges faster than SGD.

Then we consider strongly convex functions $f_{j,k}(x) = \exp(x_j - k) + \exp(k - x_j) + \frac{1}{2}||x||^2$ for $(i,k) \in \mathcal{E}$ and

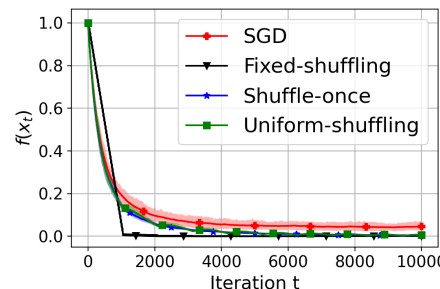

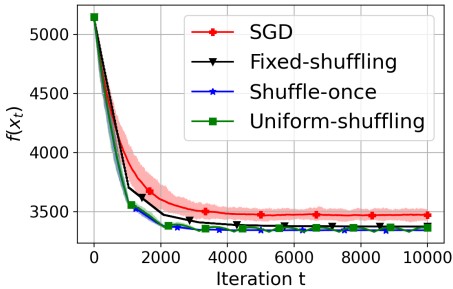

Figure 1: Experimental Results on Convex (up) and Strongly-convex (down) Objective Functions.

their sample average below.

$$f(x) = \frac{1}{1050}\sum_{(k,i)\in\mathcal{E}} f_{i,k}(x) = \frac{1}{2}||x||^2 +$$

$$\frac{\exp(n+1) - \exp(-n)}{1050[\exp(1)-1]}\sum_{j=1}^{50}[\exp(x_j) + \exp(-x_j)].$$

All these functions $f_{j,k}$ and $f$ are 1-strongly convex and $\ell$-smooth (with $\ell(u) = 5u + 5$) but not Lipschitz-smooth. We repeat the experiment in the same procedure above, except that all the stepsizes are fine-tuned to be $10^{-5}$. The result is shown in the right of Figure 1, which also shows that Algorithm 1 with all shuffling schemes converges faster than SGD.

### 5.2. Application to Phase Retrieval and DRO

We compare SGD with Algorithm 1 on phase retrieval and distributionally robust optimization (DRO), which are $\ell$-smooth but not Lipschitz smooth. We use similar setup as in (Chen et al., 2023).

In the phase retrieval problem (2), we select $m = 3000$ and $d = 100$, and generate independent Gaussian variables $x, a_r \sim \mathcal{N}(0, 0.5I_d)$, initialization $z_0 \sim \mathcal{N}(5, 0.5I_d)$, as well as $y_i = |a_r^\top z|^2 + n_i$ with noise $n_i \sim \mathcal{N}(0, 4^2)$ for $i = 1, \ldots, m$. We select constant stepsizes $2 \times 10^{-6}$ and $\eta_j^{(t)} \equiv \frac{0.007}{m}$ for SGD and Algorithm 1 respectively by fine-tuning and implement each algorithm 100 times. For Algorithm 1,

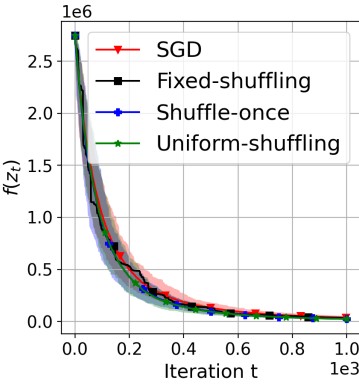

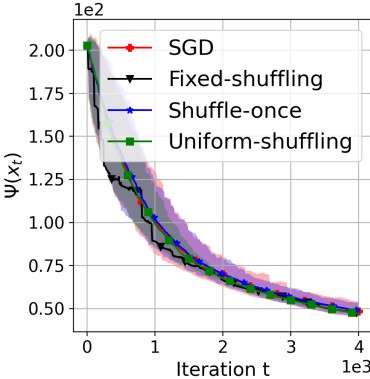

Figure 2: Experimental Results on Phase Retrieval (up) and DRO (down).

we adopt three shuffling schemes as elaborated in Section 3. The fixed-shuffling scheme and shuffling-once fix all permutations $\pi^{(t)}$ respectively to be the natural sequence $1, 2, \ldots, 3000$ and its random permutation at the beginning, while the uniform-shuffling scheme obtains permutations $\pi^{(t)}$ uniformly at random and independently for all iterations $t$. We plot the learning curves of the objective function values averaged among the 100 times, as well as the 95% and 5% percentiles in the left of Figure 2, which shows that Algorithm 1 with shuffle-once and uniform-shuffling schemes converge faster than SGD.

In the DRO problem (1), we select $\lambda = 0.01$ and $\psi^*(t) = \frac{1}{4}(t+2)_+^2 - 1$ (corresponding to $\psi$ being $\chi^2$ divergence). For the stochastic samples $\xi$, we use the life expectancy data[1] designed for regression task between the life expectancy (target) and its factors (features) of 2413 people, and preprocess the data by filling the missing values with the median of the corresponding features, censorizing and normaliz-

[1] https://www.kaggle.com/datasets/kumarajarshi/life-expectancy-who?resource=download

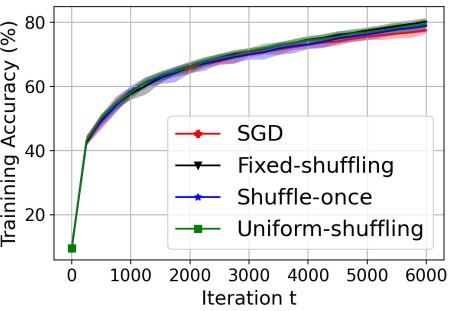

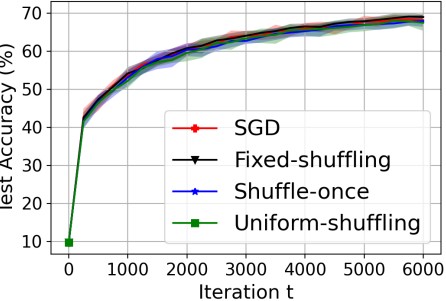

Figure 3: Experimental Results on Cifar 10 Dataset.

ing all the features [2], removing two categorical features ("country" and "status"), and adding standard Gaussian noise to the target to get robust model. We use the first 2000 samples $\{x_i, y_i\}_{i=1}^{2000}$ with features $x_i \in \mathbb{R}^{34}$ and targets $y_i \in \mathbb{R}$ for training. We use the loss function $\ell_\xi(w) = \frac{1}{2}(y_\xi - x_\xi^\top w)^2 + 0.1 \sum_{j=1}^{34} \ln\left(1 + |w^{(j)}|\right)$ of $w = [w^{(1)}; \ldots; w^{(34)}] \in \mathbb{R}^{34}$ for any sample $x_\xi, y_\xi$. We use initialization $\eta_0 = 0.1$ and $w_0 \in \mathbb{R}^{34}$ from standard Gaussian distribution.

Then similar to phase retrieval, we implement both SGD and the three sampling schemes of Algorithm 1 100 times with stepsizes $\eta_j^{(t)} = \frac{\eta_t}{n} = 10^{-7}$. We evaluate $\Psi(x_t) := \min_{\eta \in \mathbb{R}} L(x_t, \eta)$ every 10 iterations. The average, 5% and 95% percentiles of $\Psi(x_t)$ among the 100 implementations are plotted in the right of Figure 2, which shows that Algorithm 1 with fixed shuffling converges faster than SGD.

### 5.3. Application to Image Classification

We train Resnet18 (He et al., 2016) with cross-entropy loss for image classification task on Cifar 10 dataset (Krizhevsky, 2009), using SGD and Algorithm 1 with three shuffling schemes. We implement each algorithm 100 times with batchsize 200 and stepsize $10^{-3}$. After every 250 iterations, we evaluate the sample-average loss value as well as clas-

[2] The detailed process of filling missing values and censorization: https://thecleverprogrammer.com/2021/01/06/life-expectancy-analysis-with-python/

sification accuracy on the whole training dataset and test dataset. The average, 5% and 95% percentiles of these evaluated metrics among the 100 implementations are plotted in Figure 3, which shows that Algorithm 1 with fixed-shuffling scheme outperforms SGD on both training and test data, and Algorithm 1 with the other two shuffling schemes outperforms SGD on training data.

## 6. Conclusion

We revisited the convergence of shuffling-type gradient algorithms under relaxed smoothness assumptions, establishing their convergence for nonconvex, strongly convex, and non-strongly convex settings. By introducing a more general smoothness condition, we demonstrated that these methods achieve competitive convergence rates without requiring Lipschitz smoothness, extending their applicability to a broader range of optimization problems. Our analysis covers both random reshuffling and arbitrary shuffling schemes, showing that properly chosen step sizes can ensure efficient convergence in both cases. Numerical experiments further validate our theoretical findings, demonstrating that shuffling-type methods outperform SGD in non-Lipschitz scenarios. These results provide a foundation for future work on shuffling-based optimization, including its integration with variance reduction techniques, possible tighter bounds in certain situations and its application to large-scale machine learning problems.

## Impact Statement

This paper presents work whose goal is to advance the field of Machine Learning. There are many potential societal consequences of our work, none which we feel must be specifically highlighted here.

## Acknowledgments

This work was partially supported by NSF IIS 2347592, 2348169, DBI 2405416, CCF 2348306, CNS 2347617.

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

# A. Appendix / supplemental material

## A.1. Nonconvex Case Analysis

In this section we prove the theorems in section 4.1.1.

### A.1.1. LEMMAS

In this part we use notations as defined in Theorem 4.4, for completeness we repeat them here:

$$H := \frac{4\Delta_1}{\delta}, G := \sup\{u \geq 0 | u^2 \leq 2\ell(2u) \cdot H\} < \infty,$$

$$G' := \sqrt{2(1 + nA)}G + \sqrt{2n}\sigma, L := \ell(2G').$$

We first state some lemmas that are useful in our proof. The following lemma is a natural corollary of Definition 2.2, by the fact that $\ell$ is non-decreasing.

**Lemma A.1.** *If $F$ is $\ell$-smooth, for any $w \in \mathrm{dom}(F)$ satisfying $\|\nabla F(w)\| \leq G$, we have $\mathcal{B}(w, G/\ell(2G)) \subseteq \mathrm{dom}(F)$. For any $w_1, w_2 \in \mathcal{B}(w, G/\ell(2G)))$,*

$$\|\nabla F(w_1) - \nabla F(w_2)\| \leq \ell(2G)\|w_1 - w_2\|,$$

$$F(w_1) \leq F(w_2) + \langle \nabla F(w_2), w_1 - w_2 \rangle + \frac{\ell(2G)}{2}\|w_1 - w_2\|^2.$$

The following lemma gives relationship between $\|\nabla f(w; i)\|$ and $\|\nabla F(w)\|$.

**Lemma A.2.** *If Assumption 4.3 is true, we have*

$$\|\nabla f(w; i)\| \leq \sqrt{2(1 + nA)}\|\nabla F(w)\| + \sqrt{2n}\sigma.$$

*Proof.* From Assumption 4.3 we have that

$$\begin{aligned}
\|\nabla f(w; i)\|^2 &\leq 2\|\nabla f(w; i) - \nabla F(w)\|^2 + 2\|\nabla F(w)\|^2 \\
&\leq 2\sum_{i=1}^{n} \|\nabla f(w; i) - \nabla F(w)\|^2 + 2\|\nabla F(w)\|^2 \\
&\leq 2nA\|\nabla F(w)\|^2 + 2n\sigma^2 + 2\|\nabla F(w)\|^2 \\
&= 2(1 + nA)\|\nabla F(w)\|^2 + 2n\sigma^2.
\end{aligned}$$

Taking square root on both sides and notice that $\|\nabla F(w)\| \geq 0, \sigma > 0$ we have the conclusion. $\square$

According to Lemma A.2, for $w$ such that $\|\nabla F(w)\| \leq G$ is true, we have

$$\|\nabla f(w; i)\| \leq \sqrt{2(1 + nA)}G + \sqrt{2n}\sigma = G'$$

holds for all $i \in [n]$.

In our proof, we want that with high probability, $L$-Lipschitz smoothness in Lemma A.1, for both $F(w)$ and $f(w; i)$, between $w_0^{(t)}$ and $w_j^{(t)}$ is true, for $t \in [T]$, $i, j \in [n]$. For that purpose, we can prove the following inequalities with high probability, for $t \in [T]$:

$$\begin{aligned}
&\|\nabla F(w_0^{(t)})\| \leq G; \\
&\|w_0^{(t)} - w_n^{(t)}\| \leq G'/\ell(G + G'); \\
&\|\nabla f(w_0^{(t)}; i)\| \leq G', \forall i \in [n]; \\
&\|w_0^{(t)} - w_j^{(t)}\| \leq G'/\ell(2G'), \forall j \in [n].
\end{aligned} \tag{4}$$

By Lemma A.2 we know that the third inequality in (4) holds if the first inequality is true. Noticing that $\ell(G + G') \le \ell(2G')$, it suffices to prove that, for $t \in [T]$,

$$\|\nabla F(w_0^{(t)})\| \le G, \|w_0^{(t)} - w_j^{(t)}\| \le G'/\ell(2G'), \forall j \in [n]. \tag{5}$$

For the first inequality, it can be hard to bound the gradient norm directly. The following lemma states the connection between gradient norm and function value of an $\ell$-smooth function.

**Lemma A.3.** *(Lemma 3.5 in Li et al. (2023a)) If $F$ is $\ell$-smooth, then*

$$\|\nabla F(w)\|^2 \le 2\ell(2\|\nabla F(w)\|) \cdot (F(w) - F^*)$$

*for any $w \in \text{dom}(F)$.*

Since $\ell$ is sub-quadratic, with Lemma A.3 we can bound the gradient norm by bounding the difference between the function value and the optimal value. To ease the proof, let us define the following stopping time:

$$\tau := \min\{t | F(w_0^{(t)}) - F^* > H\} \wedge (T + 1).$$

For $t < \tau$, we have $\|\nabla F(w_0^{(t)})\| \le G$ based on the definition of $\tau$ and Lemma A.3, so the first inequality in (5) is satisfied. The following lemma proves that the other inequality in (5) is true for $t < \tau$ as well, therefore guarantees the Lipschitz smoothness before $\tau$.

**Lemma A.4.** *For $t < \tau$, $\eta_t \le \frac{1}{2L}$, we have for all $k \in [n]$ and $t \in [T]$, $\|w_0^{(t)} - w_k^{(t)}\|^2 \le G'/\ell(2G')$.*

*Proof.* We use induction to prove that

$$w_j^{(t)} \in \mathcal{B}(w_0^{(t)}, \frac{G'}{\ell(2G')}), j = 0, 1, \ldots, n.$$

First of all, this claim is true for $j = 0$. Now suppose the claim is true for $j \le k - 1$, i.e.,

$$\|w_0^{(t)} - w_j^{(t)}\| \le \frac{G'}{\ell(2G')}, j = 0, 1, \ldots, k - 1,$$

we try to prove it for $w_k^{(t)}$. From Lemma A.1, we have Lipschitz smoothness, for all $f(w; i)$, between $w_0^{(t)}$ and $w_j^{(t)}$, if $j \le k - 1$.

Since we have

$$\|\nabla f(w_0^{(t)}; i)\| \le G', \forall i \in [n],$$

for any $i \in [n]$ and $j \in [k - 1]$ we have

$$\|\nabla f(w_j^{(t)}; i)\| \le \|\nabla f(w_0^{(t)}; i)\| + \|\nabla f(w_j^{(t)}; i) - \nabla f(w_0^{(t)}; i)\| \le G' + L\|w_j^{(t)} - w_0^{(t)}\| \le 2G'.$$

Hence, by the algorithm design we have

$$\|w_k^{(t)} - w_0^{(t)}\| = \left\|\sum_{j=0}^{k-1} \frac{\eta_t}{n} \nabla f(w_j^{(t)}; \pi_j^{(t)})\right\| \le \sum_{j=0}^{k-1} \frac{\eta_t}{n} \|\nabla f(w_j^{(t)}; \pi_j^{(t)})\| \le \sum_{j=0}^{k-1} \frac{2G'\eta_t}{n} \le \frac{G'}{L} = \frac{G'}{\ell(2G')},$$

where the third inequality uses $k \le n$ and $\eta_t \le \frac{1}{2L}$. By induction, the claim is true. $\square$

Therefore, we have the desired Lipschitz smoothness property in Lemma A.1 for $t < \tau$. Our next target is to bound $\mathbb{P}(\tau \le T)$.

To simplify the notations, let us define

$$\epsilon_k^{(t)} := \frac{1}{n} \sum_{j=0}^{k-1} (\nabla f(w_j^{(t)}; \pi_{j+1}^{(t)}) - \nabla f(w_0^{(t)}; \pi_{j+1}^{(t)}))$$

as the average of differences between the gradients at the start of iteration $t$ and the actual gradients we used until step $j$ in the $t$-th outer iteration. It is worth mentioning that the actual step in $t$-th outer iteration is $-\eta_t[\nabla F(w_0^{(t)}) + \epsilon_n^{(t)}]$.

Now we bound the probability of event $\{\tau \leq T\}$ by bounding the expectation of function value at the stopping time.

**Lemma A.5.** *With parameters chosen in Theorem 4.4, we have*

$$\mathbb{E}[F(w_0^{(\tau)}) - F^*] \leq 2\Delta_1.$$

*Proof.* For any $t < \tau$,

$$
\begin{aligned}
&F(w_0^{(t+1)}) - F(w_0^{(t)}) \\
&\leq \langle \nabla F(w_0^{(t)}), w_0^{(t+1)} - w_0^{(t)} \rangle + \frac{L}{2}\|w_0^{(t+1)} - w_0^{(t)}\|^2 \\
&= -\eta_t \langle \nabla F(w_0^{(t)}), \nabla F(w_0^{(t)}) + \epsilon_n^{(t)} \rangle + \frac{L\eta_t^2}{2}\|\nabla F(w_0^{(t)}) + \epsilon_n^{(t)}\|^2 \\
&= -\frac{\eta_t}{2}(\|\nabla F(w_0^{(t)})\|^2 + \|\nabla F(w_0^{(t)}) + \epsilon_n^{(t)}\|^2 - \|\epsilon_n^{(t)}\|^2) + \frac{L\eta_t^2}{2}\|\nabla F(w_0^{(t)}) + \epsilon_n^{(t)}\|^2 \\
&\leq -\frac{\eta_t}{2}\|\nabla F(w_0^{(t)})\|^2 + \frac{\eta_t}{2}\|\epsilon_n^{(t)}\|^2 \\
&\leq -\frac{\eta_t}{2}\|\nabla F(w_0^{(t)})\|^2 + \frac{\eta_t L^2}{2n}\sum_{k=0}^{n-1}\|w_k^{(t)} - w_0^{(t)}\|^2.
\end{aligned}
\tag{6}
$$

Here the first and last inequalities are from Lemma A.1 and the second is because $\eta_t \leq \frac{1}{2L}$. Taking summation from $t = 1$ to $t = \tau - 1$ and taking expectation we have

$$\mathbb{E}[F(w_0^{(\tau)}) - F^*] \leq \Delta_1 - \mathbb{E}[\sum_{t=1}^{\tau-1}\frac{\eta_t}{2}\|\nabla F(w_0^{(t)})\|^2] + \mathbb{E}[\sum_{t=1}^{\tau-1}\frac{\eta_t L^2}{2n}\sum_{k=0}^{n-1}\|w_k^{(t)} - w_0^{(t)}\|^2]. \tag{7}$$

Now let us get a bound for the last term on the right hand side. For any $t \in [T]$, $k \in [n]$, from Algorithm 1 and Cauchy-Schwarz inequality we have

$$
\begin{aligned}
\|w_k^{(t)} - w_0^{(t)}\|^2 &= \frac{k^2\eta_t^2}{n^2}\left\|\frac{1}{k}\sum_{j=0}^{k-1}\nabla f(w_j^{(t)}; \pi_{j+1}^{(t)})\right\|^2 \\
&\leq \frac{3k^2\eta_t^2}{n^2}\|\frac{1}{k}\sum_{j=0}^{k-1}(\nabla f(w_0^{(t)}; \pi_{j+1}^{(t)}) - \nabla F(w_0^{(t)}))\|^2 + \frac{3k^2\eta_t^2}{n^2}\|\nabla F(w_0^{(t)})\|^2 \\
&\quad + \frac{3k\eta_t^2}{n^2}\sum_{j=0}^{k-1}\|\nabla f(w_j^{(t)}; \pi_{j+1}^{(t)}) - \nabla f(w_0^{(t)}; \pi_{j+1}^{(t)})\|^2.
\end{aligned}
$$

Let us denote the 3 terms on the RHS as $A_1(t, k)$, $A_2(t, k)$ and $A_3(t, k)$, i.e. $\|w_k^{(t)} - w_0^{(t)}\|^2 \leq A_1(t, k) + A_2(t, k) + A_3(t, k)$. Since we are interested in $\mathbb{E}[\sum_{t=1}^{\tau-1}\frac{\eta_t L^2}{2n}\sum_{k=0}^{n-1}\|w_k^{(t)} - w_0^{(t)}\|^2]$, we need to bound $\mathbb{E}[\sum_{t=0}^{\tau-1}\frac{\eta_t L^2}{2n}\sum_{k=1}^{n-1}A_i(t, k)]$ for $i = 1, 2, 3$.

For $A_1(t, k)$, since $\pi^{(t)}$ is randomly chosen, let $\mathcal{F}_t := \sigma(\pi^{(1)}, \cdots, \pi^{(t)})$ be the $\sigma$-algebra generated in Algorithm 1, for

$t \in [T]$ we have

$$\mathbb{E}[\frac{\eta_t L^2}{2n} \sum_{k=0}^{n-1} A_1(t,k)|\mathcal{F}_{t-1}] = \frac{\eta_t L^2}{2n} \sum_{k=0}^{n-1} \frac{3k^2\eta_t^2}{n^2} \mathbb{E}[\|\frac{1}{k}\sum_{j=0}^{k-1} \nabla f(w_0^{(t)}; \pi_{j+1}^{(t)}) - \nabla F(w_0^{(t)})\|^2|\mathcal{F}_{t-1}]$$

$$= \frac{\eta_t L^2}{2n} \sum_{k=0}^{n-1} \frac{3k^2\eta_t^2}{n^2} \frac{n-k}{k(n-1)} \frac{1}{n} \sum_{i=0}^{n-1} \|\nabla f(w_0^{(t)}; i+1) - \nabla F(w_0^{(t)})\|^2$$

$$\leq \frac{\eta_t L^2}{2n} \sum_{k=0}^{n-1} \frac{3\eta_t^2 k(n-k)}{n^2(n-1)} (A\|\nabla F(w_0^{(t)})\|^2 + \sigma^2)$$

$$\leq \frac{\eta_t^3 L^2}{2n} (A\|\nabla F(w_0^{(t)})\|^2 + \sigma^2).$$

Here the second equation comes from variance of randomized reshuffling variables, (Lemma 1 in Mishchenko et al. (2020)); the first inequality is from assumption 4.3; the last inequality is because $\sum_{k=0}^{n-1} k(n-k) = \frac{(n-1)n(n+1)}{6} \leq \frac{n^2(n-1)}{3}$.

Let $\{Z_t\}_{t \leq T}$ be a sequence such that $Z_1 = 0$ and for any $t \in [2, T]$,

$$Z_t - Z_{t-1} = -\frac{\eta_t^3 L^2}{2n} (A\|\nabla F(w_0^{(t)})\|^2 + \sigma^2) + \frac{\eta_t L^2}{2n} \sum_{k=0}^{n-1} A_1(t-1,k).$$

We know $\{Z_t\}$ is a supermartingale. Since $\tau$ is a bounded stopping time, by optional stopping theorem, we have $\mathbb{E}[Z_\tau] \leq \mathbb{E}[Z_1]$, which leads to

$$\mathbb{E}[\sum_{t=1}^{\tau-1} \frac{\eta_t L^2}{2n} \sum_{k=0}^{n-1} A_1(t,k)] \leq \mathbb{E}[\sum_{t=1}^{\tau-1} \frac{\eta_t^3 L^2}{2n} (A\|\nabla F(w_0^{(t)})\|^2 + \sigma^2)].$$

For $A_2(t,k)$, for any $t \in [T]$, taking summation over $k$ we have $\sum_{k=0}^{n-1} A_2(t,k) \leq n\eta_t^2 \|\nabla F(w_0^{(t)})\|^2$, therefore

$$\mathbb{E}[\sum_{t=1}^{\tau-1} \frac{\eta_t L^2}{2n} \sum_{k=0}^{n-1} A_2(t,k)] \leq \mathbb{E}[\sum_{t=1}^{\tau-1} \frac{\eta_t^3 L^2}{2} \|\nabla F(w_0^{(t)})\|^2].$$

For $A_3(t,k)$, for any $t < \tau$, by Lemma A.1 we have

$$A_3(t,k) \leq \frac{3kL^2\eta_t^2}{n^2} \sum_{j=0}^{n-1} \|w_j^{(t)} - w_0^{(t)}\|^2.$$

Taking summation over $k$, taking expectation we have

$$\mathbb{E}[\sum_{t=1}^{\tau-1} \frac{\eta_t L^2}{2n} \sum_{k=0}^{n-1} A_3(t,k)] \leq \mathbb{E}[\sum_{t=1}^{\tau-1} \frac{3\eta_t^3 L^4}{4n} \sum_{j=0}^{n-1} \|w_j^{(t)} - w_0^{(t)}\|^2].$$

Now putting these together, we have

$$\mathbb{E}[\sum_{t=1}^{\tau-1} \frac{\eta_t L^2}{2n} \sum_{j=0}^{n-1} \|w_j^{(t)} - w_0^{(t)}\|^2] \leq \mathbb{E}[\sum_{t=1}^{\tau-1} \frac{\eta_t^3 L^2}{2n} (A\|\nabla F(w_0^{(t)})\|^2 + \sigma^2)] + \mathbb{E}[\sum_{t=1}^{\tau-1} \frac{\eta_t^3 L^2}{2} \|\nabla F(w_0^{(t)})\|^2]$$

$$+ [\sum_{t=1}^{\tau-1} \frac{3\eta_t^3 L^4}{4n} \sum_{j=0}^{n-1} \|w_j^{(t)} - w_0^{(t)}\|^2].$$

Since $\eta_t \leq \frac{1}{2L} < \frac{1}{\sqrt{3}L}$ we have $\frac{3\eta_t^3 L^4}{4n} \leq \frac{\eta_t L^2}{4n}$, rearranging the terms we have

$$\mathbb{E}[\sum_{t=1}^{\tau-1} \sum_{j=0}^{n-1} \|w_j^{(t)} - w_0^{(t)}\|^2] \leq \mathbb{E}[\sum_{t=1}^{\tau-1} 2\eta_t^2 \sigma^2] + 2n\mathbb{E}[\sum_{t=1}^{\tau-1} \eta_t^2(\frac{A}{n} + 1)\|\nabla F(w_0^{(t)})\|^2]. \tag{8}$$

Put this into (7) we have,

$$
\mathbb{E}[F(w_0^{(\tau)}) - F^*]
$$

$$
\leq \Delta_1 + \mathbb{E}\Big[\sum_{t=1}^{\tau-1}\Big( -\frac{\eta_t}{2}\|\nabla F(w_0^{(t)})\|^2 + \frac{\eta_t L^2}{2n}\sum_{j=0}^{n-1}\|w_j^{(t)} - w_0^{(t)}\|^2 \Big)\Big]
$$

$$
\overset{(8)}{\leq} \Delta_1 + \mathbb{E}\Big[\sum_{t=1}^{\tau-1}\frac{L^2\sigma^2\eta_t^3}{n} - \sum_{t=1}^{\tau-1}\Big(\big(\frac{\eta_t}{2} - (\frac{A}{n}+1)\eta_t^3 L^2\big)\|\nabla F(w_0^{(t)})\|^2\Big)\Big]
$$

$$
\leq \Delta_1 + \mathbb{E}\Big[\sum_{t=1}^{\tau-1}\frac{L^2\sigma^2\eta_t^3}{n} - \sum_{t=1}^{\tau-1}\Big(\frac{\eta_t}{4}\|\nabla F(w_0^{(t)})\|^2\Big)\Big] \qquad (9)
$$

$$
\leq \Delta_1 + \frac{L^2\sigma^2}{n}\sum_{t=1}^{T}\eta_t^3.
$$

Here the third inequality is from $\eta_t \leq \frac{1}{2L\sqrt{\frac{A}{n}+1}}$ and the last inequality is because $\eta_t > 0$ and $\tau \leq T+1$.

Since $\sum_{t=1}^{T}\eta_t^3 \leq \frac{n\Delta_1}{\sigma^2 L^2}$, we have $\mathbb{E}[F(w_0^{(\tau)}) - F^*] \leq 2\Delta_1$. $\qquad\square$

Now we can bound the probability that $\tau = T+1$.

**Lemma A.6.** *With the parameters in Theorem 4.4, we have*

$$
\mathbb{P}(\tau \leq T) \leq \delta/2.
$$

*Proof.* From Lemma A.5 and the value of $H$ we have

$$
\mathbb{P}(\tau \leq T) \leq \mathbb{P}(F(w_0^{(\tau)}) - F^* > H) \leq \frac{\mathbb{E}[F(w_0^{(\tau)}) - F^*]}{H} \leq \frac{2\Delta_1}{H} = \frac{\delta}{2}.
$$

$\qquad\square$

### A.1.2. PROOF FOR THEOREMS IN NONCONVEX CASES

**Proof for Theorem 4.4**

*Proof.* From (9) we have

$$
\mathbb{E}[F(w_0^{\tau}) - F^*] + \mathbb{E}\Big[\sum_{t=1}^{\tau-1}\frac{\eta_t}{4}\|\nabla F(w_0^{(t)})\|^2\Big] \leq \Delta_1 + \frac{L^2\sigma^2}{n}\sum_{t=1}^{T}\eta_t^3 \leq 2\Delta_1. \qquad (10)
$$

Therefore, since $\delta \leq 1$ we have

$$
\frac{8\Delta_1}{\eta_T} \geq \mathbb{E}\Big[\sum_{t=1}^{\tau-1}\|\nabla F(w_0^{(t)})\|^2\Big]
$$

$$
\geq \mathbb{P}(\tau = T+1)\mathbb{E}\Big[\sum_{t=1}^{T}\|\nabla F(w_0^{(t)})\|^2|\tau = T+1\Big]
$$

$$
\geq \frac{1}{2}\mathbb{E}\Big[\sum_{t=1}^{T}\|\nabla F(w_0^{(t)})\|^2|\tau = T+1\Big].
$$

By Markov's inequality and our choice of $T$, we have

$$
\mathbb{P}\Big(\frac{1}{T}\sum_{t=1}^{T}\|\nabla F(w_0^{(t)})\|^2 > \epsilon^2|\tau = T+1\Big) \leq \frac{16\Delta_1}{\eta_T T\epsilon^2} \leq \frac{\delta}{2}.
$$

From Lemma ([A.6](#)) we have $\mathbb{P}(\tau \leq T) \leq \frac{\delta}{2}$. Therefore,

$$\mathbb{P}\Big(\{\frac{1}{T}\sum_{t=1}^{T}\|\nabla F(w_0^{(t)})\|^2 > \epsilon^2\} \cup \{\tau \leq T\}\Big)$$

$$\leq \mathbb{P}(\tau \leq T) + \mathbb{P}\Big(\frac{1}{T}\sum_{t=1}^{T}\|\nabla F(w_0^{(t)})\|^2 > \epsilon^2 | \tau = T+1\Big)$$

$$\leq \frac{\delta}{2} + \frac{\delta}{2} = \delta.$$

Since $L = \ell(2G') = \Omega(G'^p) = \Omega(n^{\frac{p}{2}})$, with $\eta = \mathcal{O}(\sqrt[3]{\frac{n^{1-p}}{T}})$ and $T = \mathcal{O}(\frac{n^{\frac{p-1}{2}}}{\epsilon^3})$ we have the complexity. $\qquad\square$

The following lemma is useful in the proof of arbitrary scheme.

**Lemma A.7.** *(lemma 6 in ([Nguyen et al., 2021](#))) For $t < \tau$ and $0 < \eta_t \leq \frac{1}{L\sqrt{3}}$, we have*

$$\sum_{j=0}^{n-1}\|w_j^{(t)} - w_0^{(t)}\|^2 \leq n\eta_t^2[(3A+2)\|\nabla F(w_0^{(t)})\|^2 + 3\sigma^2].$$

**Proof for Theorem [4.6](#)**

*Proof.* From inequality ([6](#)) we have for any $t < \tau$,

$$F(w_0^{(t+1)}) - F(w_0^{(t)})$$

$$\leq -\frac{\eta_t}{2}\|\nabla F(w_0^{(t)})\|^2 + \frac{\eta_t L^2}{2n}\sum_{k=0}^{n-1}\|w_k^{(t)} - w_0^{(t)}\|^2$$

$$\leq -\frac{\eta_t}{2}\|\nabla F(w_0^{(t)})\|^2 + \frac{\eta_t^3 L^2[(3A+2)\|\nabla F(w_0^{(t)})\|^2 + 3\sigma^2]}{2}$$

$$\leq -\frac{\eta_t}{4}\|\nabla F(w_0^{(t)})\|^2 + \frac{3\eta_t^3 L^2\sigma^2}{2},$$

where the second inequality is from Lemma [A.7](#) and the last inequality is from $\eta_t \leq \frac{1}{L\sqrt{2(3A+2)}}$. Now taking summation of $t$ from 1 to $\tau - 1$ we have

$$F(w_0^{(\tau)}) - F^* \leq F(w_0^{(\tau)}) - F^* + \sum_{t=1}^{\tau-1}\frac{\eta_t}{4}\|\nabla F(w_0^{(t)})\|^2 \leq \Delta_1 + \frac{3L^2\sigma^2}{2}\sum_{t=1}^{\tau-1}\eta_t^3 \leq 2\Delta_1,$$

where the last inequality is because $\tau \leq T + 1$ and the choice of $\eta_t$. Therefore we have $\tau = T + 1$ since $H \geq 2\Delta_1$. On the other hand, we also have

$$\frac{8\Delta_1}{\eta_T} \geq \sum_{t=1}^{\tau-1}\|\nabla F(w_0^{(t)})\|^2$$

$$= \sum_{t=1}^{T}\|\nabla F(w_0^{(t)})\|^2.$$

Therefore, we have

$$\frac{1}{T}\sum_{t=1}^{T}\|\nabla F(w_0^{(t)})\|^2 \leq \frac{8\Delta_1}{T\eta_T} \leq \epsilon^2$$

from our choice of $T$. $\qquad\square$

## A.2. Strongly Convex Case Analysis

**Lemma A.8.** *If we let $H \geq \frac{3\sigma^2}{4\mu} \log(\frac{4}{\epsilon}) + \Delta_1$ and $\eta_t = \eta$, we have $\tau \geq \frac{2}{\mu\eta} \log(\frac{4}{\epsilon})$.*

*Proof.* From inequality (6) we have for $t < \tau$

$$F(w_0^{(t+1)}) - F(w_0^{(t)}) \leq -\frac{\eta}{2}\|\nabla F(w_0^{(t)})\|^2 + \frac{\eta L^2}{2n} \sum_{k=0}^{n-1} \|w_k^{(t)} - w_0^{(t)}\|^2$$

$$\leq -\frac{\eta}{2}\|\nabla F(w_0^{(t)})\|^2 + \frac{\eta^3 L^2[(3A+2)\|\nabla F(w_0^{(t)})\|^2 + 3\sigma^2]}{2}$$

$$\leq \frac{3\eta\sigma^2}{8},$$

where the last inequality is from $\eta \leq \frac{1}{L\sqrt{2(3A+2)}} \leq \frac{1}{2L}$. From the definition of $\tau$ we have

$$\tau \geq 1 + \frac{8(H - \Delta_1)}{3\eta\sigma^2} \geq \frac{2}{\mu\eta} \log(\frac{4}{\epsilon}).$$

$\square$

## Proof for Theorem 4.8

*Proof.* From Lemma A.6 and the parameter choices we have $\mathbb{P}(\tau \leq T) \leq \frac{\delta}{2}$.

Now we try to bound $F(w_0^{(\tau)}) - F^*$. In the strongly convex case, for $t < \tau$ we have

$$F(w_0^{(t+1)}) \leq F(w_0^{(t)}) - \frac{\eta_t}{2}\|\nabla F(w_0^{(t)})\|^2 + \frac{L^2 \eta_t}{2n} \sum_{j=0}^{n-1} \|w_j^{(t)} - w_0^{(t)}\|^2$$

$$\leq F(w_0^{(t)}) - \frac{\mu\eta_t}{2}(F(w_0^{(t)}) - F^*) - \frac{\eta_t}{4}\|\nabla F(w_0^{(t)})\|^2 + \frac{L^2 \eta_t}{2n} \sum_{j=0}^{n-1} \|w_j^{(t)} - w_0^{(t)}\|^2,$$

here the first inequality is from (6) and the second one is from strongly convexity. We can rearrange the items and write the above inequality as

$$F(w_0^{(t+1)}) - F^* \leq \left(1 - \frac{\mu\eta_t}{2}\right)(F(w_0^{(t)}) - F^*) + \frac{L^2\sigma^2\eta_t^3}{n} + A(t), \tag{11}$$

where $A(t)$ is defined as

$$A(t) := \frac{L^2 \eta_t}{2n} \sum_{j=0}^{n-1} \|w_j^{(t)} - w_0^{(t)}\|^2 - \frac{\eta_t}{4}\|\nabla F(w_0^{(t)})\|^2 - \frac{L^2\sigma^2\eta_t^3}{n}. \tag{12}$$

Let $\eta_t = \eta := \frac{4\log(\sqrt{n}T)}{\mu T}$, we want $1 - \frac{\mu\eta}{2} > 0$, therefore we need $\frac{T}{\log(\sqrt{n}T)} \geq 2$. Taking expectation and summation we have

$$\mathbb{E}[F(w_0^{(\tau)}) - F^*] \leq \mathbb{E}[(1 - \frac{\mu\eta}{2})^{\tau-1}\Delta_1] + \frac{2L^2\sigma^2\eta^2}{n\mu}[1 - (1 - \frac{\mu\eta}{2})^{\tau-1}]$$

$$+ \mathbb{E}[\sum_{t=1}^{\tau-1}(1 - \frac{\mu\eta}{2})^{\tau-1-t}A(t)]$$

$$\leq \Delta_1\mathbb{E}[\exp(-\mu\eta\tau/2)] + \frac{2L^2\sigma^2\eta^2}{n\mu} + \mathbb{E}[\sum_{t=1}^{\tau-1}A(t)]$$

$$\leq \frac{\delta\epsilon}{8} + \frac{1}{nT^2}\left(\Delta_1 + \frac{L^2\sigma^2\log^2(\sqrt{n}T)}{\mu^3}\right) + \mathbb{E}[\sum_{t=1}^{\tau-1}A(t)],$$

where the second inequality is from $1 - x \leq \exp(-x)$ for $x \in (0, 1)$ and the last inequality is from Lemma A.8, $\mathbb{P}(\tau \leq T) \leq \delta/2$ and the value of $\eta$. Now if we look at the last item, we can notice from (8), by using $\eta \leq \frac{1}{L\sqrt{2(3A+2)}} \leq \frac{1}{2L\sqrt{\frac{A}{n}+1}}$, that we already have

$$\mathbb{E}[\sum_{t=1}^{\tau-1} A(t)] \leq 0.$$

Therefore, we have

$$\frac{\delta\epsilon}{8} + \frac{1}{nT^2}\Big(\Delta_1 + \frac{8L^2\sigma^2\log^2(\sqrt{n}T)}{\mu^3}\Big) \geq \mathbb{E}[F(w_0^{(\tau)}) - F^*]$$
$$\geq \mathbb{P}(\tau = T+1)\mathbb{E}[F(w_0^{(T+1)}) - F^*|\tau = T+1]$$
$$\geq \frac{1}{2}\mathbb{E}[F(w_0^{(T+1)}) - F^*|\tau = T+1].$$

$$\mathbb{P}(F(w_0^{(T+1)}) - F^* > \epsilon|\tau = T+1) \leq \frac{\mathbb{E}[F(w_0^{(T+1)}) - F^*|\tau = T+1]}{\epsilon}$$
$$\leq \frac{\delta}{4} + \frac{2}{\epsilon nT^2}\Big(\Delta_1 + \frac{8L^2\sigma^2\log^2(\sqrt{n}T)}{\mu^3}\Big)$$
$$\leq \frac{\delta}{4} + \frac{\delta}{8} + \frac{\delta}{8} = \frac{\delta}{2},$$

where the last line is from the constraint on T.

$\square$

### Proof for Theorem 4.9

*Proof.* The algorithm starts from $w_0^{(1)}$ and we define $S = \{w|F(w) \leq F(w_0^{(1)})\}$. Since $F$ is strongly-convex, we have $S$ being compact. Therefore, we can define $G' = \max_w\{\|\nabla f(w; i)\| \mid w \in S, i \in [n]\} < \infty$.

If we have $w_0^{(t)} \in S$ for all $t \in [T]$, we have $\|\nabla f(w_0^{(t)}; i)\| \leq G'$ for $t \in [T]$ and $i \in [n]$. On the other hand, by definition of $F$ we have $\|\nabla F(w_0^{(t)})\| \leq G'$ for $t \in [T]$. Therefore, by Lemma A.4 we have Lipschitz smoothness between $w_0^{(t)}$ and $w_j^{(t)}$, for both $F(w)$ and $f(w; i)$, for $t \in [T]$, $i, j \in [n]$. The rest of the proof then follows the one in Lipschitz smoothness case (theorem 1 in Nguyen et al. (2021)).

Now we prove that $w_0^{(t)} \in S$, for $t \in T$. The statement is obviously true for $t = 1$. Now for $t \in [2, T]$, assume that we already proved the conclusion for $1, \cdots, t-1$, we can use Lipschitz smoothness in the first $t-1$ iterations. Therefore, from theorem 1 in Nguyen et al. (2021) we have that

$$F(w_0^{(t)}) - F(w_*) \leq (1 - \rho\eta)^{t-1}\Delta_1 + \frac{D\eta^2}{\rho},$$

where $\rho = \frac{\mu}{3}$, $D = (\mu^2 + L^2)\sigma_*^2$. On the other hand, since $\eta \leq \frac{\Delta_1\rho^2}{D}$ we have

$$(1 - \rho\eta)^{t-1}\Delta_1 + \frac{D\eta^2}{\rho} \leq (1 - \rho\eta)\Delta_1 + \frac{D\eta^2}{\rho} \leq \Delta_1.$$

Therefore, we have $F(w_0^{(t)}) \leq F(w_0^{(1)})$, which means $w_0^{(t)} \in S$.

$\square$

## A.3. Non-strongly Convex Case Analysis

**Proof for theorem 4.11**

*Proof.* From Lemma A.6 we know $\mathbb{P}(\tau \leq T) < \frac{\delta}{2}$.

For $t < \tau$, if $\eta_t = \eta$, from lemma 7 in (Nguyen et al., 2021) we have that

$$\|w_0^{(t+1)} - w_*\|^2 \leq \|w_0^{(t)} - w_*\|^2 - 2\eta[F(w_0^{(t)}) - F^*] + \frac{2L\eta^3}{n^3} \sum_{i=1}^{n-1} \|\sum_{j=i}^{n-1} \nabla f(w_*; \pi_{j+1}^{(t)})\|^2, \tag{13}$$

where $w_*$ is the optimal solution. If we denote $A(t) := \sum_{i=1}^{n-1} \|\sum_{j=i}^{n-1} \nabla f(w_*; \pi_{j+1}^{(t)})\|^2$ and let $\sigma_* := \sqrt{\frac{1}{n} \sum_{i=1}^{n} \|\nabla f(w_*; i)\|^2}$, we have that for any $t \in [T]$

$$
\begin{aligned}
\mathbb{E}[A(t)] &= \sum_{i=0}^{n-1} (n-i)^2 \mathbb{E}\left[\left\|\frac{1}{n-i} \sum_{j=i}^{n-1} \nabla f(w_*; \pi_{j+1}^{(t)}) - \nabla F(w_*)\right\|^2\right] \\
&= \sum_{i=0}^{n-1} \frac{(n-i)^2 i}{n(n-i)(n-1)} \sum_{j=0}^{n-1} \|\nabla f(w_*; \pi_{j+1}^{(t)})\|^2 \\
&= \frac{n(n+1)\sigma_*^2}{6}.
\end{aligned}
$$

By optional stopping theorem we know that

$$\mathbb{E}\left[\sum_{t=1}^{\tau-1} \left(A(t) - \frac{n(n+1)\sigma_*^2}{6}\right)\right] = 0.$$

Taking summation from $t = 0$ to $\tau - 1$ for (13) and taking expectation we have

$$
\begin{aligned}
2\eta \mathbb{E}\left[\sum_{t=1}^{\tau-1}(F(w_0^{(t)}) - F^*)\right] &\leq \|w_0^{(1)} - w_*\|^2 + \frac{2L\eta^3}{n^3} \mathbb{E}\left[\sum_{t=1}^{\tau-1} \frac{n(n+1)\sigma_*^2}{6}\right] \\
&\leq \|w_0^{(1)} - w_*\|^2 + \frac{2LT\eta^3\sigma_*^2}{3n},
\end{aligned}
$$

where the second inequality uses $\tau \leq T + 1$. Therefore, we have

$$
\begin{aligned}
\frac{1}{2\eta}\left(\|w_0^{(1)} - w_*\|^2 + \frac{2LT\eta^3\sigma_*^2}{3n}\right) &\geq \mathbb{E}\left[\sum_{t=1}^{\tau-1}(F(w_0^{(t)}) - F^*)\right] \\
&\geq \frac{1}{2}\mathbb{E}\left[\sum_{t=1}^{T}(F(w_0^{(t)}) - F^*)|\tau = T + 1\right].
\end{aligned}
$$

If we define $\bar{w}_T = \frac{1}{T}\sum_{t=1}^{T} w_0^{(t)}$, from convexity we have

$$F(\bar{w}_T) - F^* \leq \frac{1}{T}\sum_{t=1}^{T}[F(w_0^{(t)}) - F^*].$$

Consider the event $\mathcal{F} := \{F(\bar{w}_T) - F^* > \epsilon\}$, we have

$$
\begin{aligned}
\mathbb{P}(\mathcal{F}|\tau = T+1) &\leq \mathbb{P}(\frac{1}{T}\sum_{t=1}^{T}(F(w_0^{(t)}) - F^*) > \epsilon|\tau = T+1) \\
&\leq \frac{\mathbb{E}\left[\sum_{t=1}^{T}(F(w_0^{(t)}) - F^*)|\tau = T+1\right]}{T\epsilon} \\
&\leq \frac{1}{\eta T\epsilon}\left(\|w_0^{(1)} - w_*\|^2 + \frac{2LT\eta^3\sigma_*^2}{3n}\right) \\
&\leq \frac{2}{\eta T\epsilon}\|w_0^{(1)} - w_*\|^2 \\
&\leq \frac{\delta}{2},
\end{aligned}
$$

where the last two inequalities are from the choices of $\eta$ and $T$, separately. $\qquad\square$

**Proof for Theorem 4.12**

*Proof.* Similar to Theorem 4.9, if we have $w_0^{(t)} \in S$ for $t \in [T]$, we have the desired Lipschitz smoothness.

Now we prove the conclusion by trying to prove that $w_0^{(t)} \in S$ for $t \in [T]$. The statement is obviously true for $t = 1$. Now for $t \in [2, T]$, assume that we already proved the conclusion for $1, \cdots, t-1$, we can use Lipschitz smoothness in the first $t-1$ iterations. Therefore, from (13) we have

$$
\|w_0^{(t)} - w_*\|^2 \leq \|w_0^{(t-1)} - w_*\|^2 - 2\eta[F(w_0^{(t-1)}) - F^*] + \frac{2L\eta^3}{n^3}\sum_{i=1}^{n-1}\|\sum_{j=i}^{n-1}\nabla f(w_*; \pi_{j+1}^{(t-1)})\|^2.
$$

If $F(w_0^{(t-1)}) - F^* \leq \epsilon$, we have the desired conclusion.

If $F(w_0^{(t-1)}) - F^* > \epsilon$, since $\eta \leq \frac{1}{G'}\sqrt{\frac{3\epsilon}{L}}$, we have

$$
\begin{aligned}
\|w_0^{(t+1)} - w_*\|^2 &\leq \|w_0^{(t)} - w_*\|^2 - 2\eta\epsilon + \frac{2L\eta^3}{n^3}\sum_{i=1}^{n-1}(n-i)^2 G'^2 \\
&\leq \|w_0^{(t)} - w_*\|^2 - 2\eta\epsilon + \frac{2L\eta^3}{n^3}\frac{n^3}{3}G'^2 \\
&\leq \|w_0^{(t)} - w_*\|^2.
\end{aligned}
$$

Therefore, if $F(w_0^{(t)}) - F^* \geq \epsilon$ for $t \in [T]$, we have $w_0^{(t)} \in S$ for $t \in [T]$. Taking summation we have that

$$
2\eta\sum_{t=1}^{T}[F(w_0^{(t)}) - F(w_*)] \leq \|w_0^{(1)} - w_*\|^2 + \frac{2LG'^2\eta^3T}{3}.
$$

Therefore we have

$$
\frac{1}{T}\sum_{t=1}^{T}[F(w_0^{(t)}) - F(w_*)] \leq \frac{1}{2\eta T}\left(\|w_0^{(1)} - w_*\|^2 + \frac{2LG'^2\eta^3T}{3}\right) \leq \epsilon.
$$

However, this contradict the assumption that $F(w_0^{(t)}) - F^* \geq \epsilon$ for $t \in [T]$. Therefore, there must be $t \in [T]$ such that $F(w_0^{(t)}) - F^* \leq \epsilon$.

$\qquad\square$

