# OpenReview forum: "Revisiting Convergence: Shuffling Complexity Beyond Lipschitz Smoothness"
_ICML.cc/2025/Conference — ICML 2025 poster_

### Official Review · Reviewer_Vaxv · 2025-02-27

**Overall Recommendation:** 4

**Summary:**

This paper establishes convergence of random reshuffling methods under a l-smoothness condition, where l is a function rather than a constant.

**Claims And Evidence:**

New results that improve our understanding of random reshuffling approaches are potentially interesting, as random reshuffling is heavily used in practice.  This paper does a good job of reviewing existing literature and placing their result in the right context. Essentially they broaden the coverage of convergence results for RR methods to a new smoothness criterion that wasn't previously covered. It is a fairly general condition and it covers several problems of interest.
In terms of impact, although this is a useful contribution to the literature it's not ground breaking, and doesn't fundamentally change our understanding of RR methods, so I would lean towards an accept but not a strong accept.

**Essential References Not Discussed:**

N/A

**Experimental Designs Or Analyses:**

Experiments seem reasonable and address the content of the paper. Error bars are shown, graphs are clear. For this sort of theory paper experiments are optional and don't need to address large scale problems.

**Methods And Evaluation Criteria:**

N/A

**Other Comments Or Suggestions:**

Please check that all uses of the term Lipschitz refer to Lipschitz smoothness, as there is a distinction between a function being Lipschitz and Lipschitz smooth. In some places (Line 80) when a function is described as non-Lipschitz the authors mean non-Lipschitz smooth.

**Other Strengths And Weaknesses:**

N/A

**Questions For Authors:**

N/A

**Relation To Broader Scientific Literature:**

See Above

**Theoretical Claims:**

I have not checked any theory from the Appendix. Results in the main body of the paper are precisely stated and make sense.

---

> ### Author Rebuttal · Authors · 2025-03-31
>
> We thank the reviewer for acknowledging our contribution and careful attention to details. We will make modifications as suggested by the reviewer.

---

### Official Review · Reviewer_VM88 · 2025-03-12

**Overall Recommendation:** 3

**Summary:**

This paper investigates shuffling-type gradient methods without assuming Lipschitz smoothness, which is often not satisfied in practice for many machine learning models. The authors propose stepsize strategies that enables convergence guarantees under a more general smoothness condition called "$\ell$-smoothness." They prove convergence rates for nonconvex, strongly convex, and general convex cases under both random reshuffling and arbitrary shuffling schemes. The paper provides theoretical contributions that match or extend existing best-known rates while requiring weaker smoothness assumptions, and validates the approach with experimental results.

**Claims And Evidence:**

The central claims about convergence rates under ℓ-smoothness are supported by theoretical analyses. While the authors demonstrate through counterexamples why Lipschitz smoothness is too restrictive in practice, the practical impact of their theoretical contribution is less clear.

Given that prior work has already established that shuffling SGD can outperform vanilla SGD, the experimental value in this paper should be focused differently. The core theoretical contribution is extending convergence guarantees to the ℓ-smoothness setting, which is more general than Lipschitz smoothness. A more meaningful experimental approach would have been to:
1. Demonstrate convergence on problems that specifically violate Lipschitz smoothness but satisfy ℓ-smoothness, which they do attempt with their DRO and phase retrieval examples, but remains unclear for the Image Classification.
2. Compare performance when using their theoretically-derived stepsizes versus other common choices to validate whether their analysis leads to practically superior parameter settings.
3. Test the limits of the approach by examining problems with varying degrees of non-Lipschitz behavior (different values of the parameter p in the ℓ-smoothness condition)

Simply showing that shuffling-type methods outperform vanilla SGD repeats what's already known rather than validating the unique aspects of this paper's theoretical contribution. The experiments would be more compelling if they directly connected to the paper's novel analytical insights rather than reestablishing the general benefits of shuffling.

**Essential References Not Discussed:**

I am not aware of any essential references that have been overlooked in this submission.

**Experimental Designs Or Analyses:**

The experimental section provides some validation of the theoretical results, except for some limitations proposed already.

**Methods And Evaluation Criteria:**

The proposed methods are theoretically sound, but the assumptions are under discussed. Notably, Assumption 4.3 (which relates the variance of component gradients to the norm of the full gradient) appears unconventional, as I have not encountered it in previous literature, and it potentially represents a strong constraint on the problem class.
The experimental findings are discussed in the Claims And Evidence section of this review.

**Other Comments Or Suggestions:**

The proof techniques would benefit from more detailed exposition in the main text, while the supplementary proofs could be refined for clarity and precision. Additionally, providing more intuitive explanations and practical discussions about why ℓ-smoothness matters in real-world applications would significantly strengthen the motivation for this work.

**Other Strengths And Weaknesses:**

Strengths:
1. The paper addresses an important gap by relaxing the Lipschitz smoothness assumption
2. The theoretical results are mathematically sound and extend previous work. In particular, the analysis involving stopping time seems to be a novel contribution.
3. The experimental results do show some benefits of shuffling-type methods

Weaknesses:
1. Except for the novel usage of stopping time, the theoretical novelty is somewhat incremental, building heavily on existing approaches
2. The dependency on 1/δ is polynomial rather than logarithmic, which is a significant limitation
3. The experimental validation doesn't sufficiently isolate the impact of the paper's specific contributions, and the connection between theory and practice is not firmly established
4. The constants G' in the arbitrary shuffling results could be impractically large

**Questions For Authors:**

Do those counterexamples satisfy (L0, L1)-smoothness? What is the motivation to further relax (L0, L1)-smoothness to $\ell$ smoothness?

Can the (theoretical/empirical) results be generalized to other common strategies, like diminishing stepsizes?

**Relation To Broader Scientific Literature:**

The work builds on existing literature in shuffling-type gradient methods and generalized smoothness conditions. While it makes connections to both areas, the novel contribution is primarily in combining these two streams of research rather than developing fundamentally new insights in either area.

**Theoretical Claims:**

I roughly checked the proof of Theorem 4.4, and it appears rigorous.

---

> ### Author Rebuttal · Authors · 2025-03-31
>
> Thanks for the careful reading and constructive feedback. Below we respond to each point in detail:
>
> **Assumption 4.3:**  Since component gradients behave as gradient estimations of the full gradient, this assumption can be viewed as a generalization of the more common assumption $\mathbb{E}[\|\nabla F(w;\xi)-\nabla F(w)\|]\leq \sigma^2$, which is used in most $\ell$-smooth work, e.g., [1], [2].
>
> **W1:**  We want to mention that another key novelty is our analysis for the condition $t<\tau$, where each step's analysis must account simultaneously for both the event $t<\tau$ and the preceding steps within the same epoch due to shuffling. This dual conditioning has not been addressed by prior work that focuses only on either shuffling or $\ell$-smoothness. Additionally, assumption 4.3 is relaxed compared to other $\ell$-smooth literature.
>
> **W2:**  Achieving a logarithmic dependency on $1/\delta$ remains an open challenge under $\ell$-smoothness, and establishing such a result would constitute a significant advancement. As noted in Remark 4.5, most existing works under $\ell$-smooth assumptions exhibit polynomial dependencies on $1/\delta$. Intuitively, under $\ell$-smoothness, the parameter $\delta$ accounts for bad cases from both variance and smoothness, whereas $L$-smooth scenarios only considers the former. Thus, significantly improving this dependency is nontrivial and would require substantial theoretical breakthroughs.
>
> **W3:**  We agree that the experiments can better support our theory following the advice in 'claim and evidence' section. However, it can be hard to implement in reality, since the function $\ell$ and constant $p$ are hard to determine. We leave it as a future work to design a better algorithm to work in practice.
>
> **W4:**  We agree with that. As we mentioned in Section 4.3, the potentially large constant $G'$ in Theorems 4.9 and 4.12 is indeed undesirable. These theorems are intended as initial steps toward a broader analysis without any variance assumptions.
>
> **Q1:** These counterexamples generally do not necessarily satisfy the $(L_0,L_1)$-smoothness conditions. One motivation to move from $(L_0,L_1)$-smoothness to generalized $\ell$-smoothness is to cover functions such as double exponential functions $f(x)=a^{b^x}$ or rational functions $f(x)=P(x)/Q(x)$, where $P$ and $Q$ are polynomials, which satisfy $(L_0,L_1)$-smoothness conditions but not $\ell$-smooth.
>
> **Q2:** Yes, we can use the theoretical results and give diminishing stepsizes. As suggested in theorem 4.4, as long as the stepsizes satisfy the constraints, they can be either constant or diminishing.
>
> [1] Li, Haochuan, Alexander Rakhlin, and Ali Jadbabaie. "Convergence of adam under relaxed assumptions." Advances in Neural Information Processing Systems 36 (2023): 52166-52196.
>
> [2] Xian, Wenhan, Ziyi Chen, and Heng Huang. "Delving into the convergence of generalized smooth minimax optimization." Forty-first International Conference on Machine Learning. 2024.

---

### Official Review · Reviewer_UN8o · 2025-03-12

**Overall Recommendation:** 3

**Summary:**

- Most of the existing literature on shuffling SGD establishes the convergence rate under traditional gradient lipschitz continuity, which is a condition that does not hold in neural networks. To address this problem, the paper studies the convergence rate of shuffling SGD under the generalized smoothness assumption.
- The paper proves the convergence rates of shuffling SGD for strongly convex, convex, and nonconvex functions, each under both random reshuffling (RR) and arbitrary shuffling schemes. The derived convergence rates recover the current best-known convergence rates established under the traditional gradient Lipschitzness assumption. Finally, the paper validates its theoretical findings by toy experiments.

---
**Update after Rebuttal**

Firstly, I apologize for not being actively engaged during the rebuttal phase. Below, I summarize my current perspective on the paper:

- My initial main concern regarding this paper was related to the technical novelty; I originally viewed the paper as combining existing techniques from shuffling SGD and general Lipschitz continuity. After carefully reviewing the authors' response and revisiting the paper, I find that the authors have addressed my concerns effectively.
- The authors' response for the dependency on $p$ in Theorems 4.9 and 4.12 fully resolves my concern. However, the current paper remains slightly confusing. Specifically, in Section 4.1 (nonconvex case), both RR and arbitrary permutation schemes employ the bounded gradient assumption. Yet, for the strongly convex and non-strongly convex cases, RR results are presented under the bounded gradient assumption, whereas results for the arbitrary permutation scheme omit this assumption. If there is no particular reason for excluding the theorems on arbitrary scheme under bounded gradient assumption, I suggest the authors include these as well.

Overall, I raise my score to 3 and lean toward acceptance.

**Claims And Evidence:**

Most claims are well-supported by theorems and propositions.

**Essential References Not Discussed:**

Essential references are appropriately cited and discussed in the paper.

**Experimental Designs Or Analyses:**

The paper includes experimental results, but they are primarily intended to verify the theoretical findings rather than serve as a main contribution. As a result, I did not thorougly review the experimental design and analysis.

**Methods And Evaluation Criteria:**

This paper is purely theoretical and does not involve empirical evaluation or benchmark datasets.

**Other Comments Or Suggestions:**

Typos:

- In line 191R, “and” is missing in the end of the line
- In line 258R, “ondom” → “on dom”
- In line 285L, “Algorithm 1 arbitrary scheme” → “Algorithm 1 with arbitrary scheme”

**Other Strengths And Weaknesses:**

Strengths

- The paper derives the convergence rate of shuffling SGD under a weaker assumption (generalized smoothness) than prior works (Lipschitz continuous gradient).
- The paper establishes the convergence rates for nonconvex, convex, and strongly-convex cases. The proofs seem correct and sound.
- The gradient assumption (Assumption 4.3) is relaxed compared to prior works on shuffling SGD.

Weakness

- The main weakness of this paper is the novelty. In my view, this paper is simply a combination of two well-established topics—shuffling SGD and generalized smoothness—without introducing new technical analysis. In my understanding, the proofs of the theorems rely on the introduction of the random variable $\tau$ to leverage local smoothness properties (just like [Li et al., 2023]), which are then applied to existing lemmas from the shuffling SGD literature. As a result, this work seems to be a straightforward application of existing results rather than a fundamentally novel contribution.
- While this paper expresses the convergence rates in terms of $n$ and $\epsilon$, recent studies on shuffling SGD ([Liu et al., 2024, Koloskova et al., 2024]) provide a more detailed characterization of convergence rates, including additional parameters such as $F(w_0)-F^*$, $\mu$ (strong convexity), $\sigma$. It would improve the completeness of the results if this paper also expressed the rates in a similarly detailed manner.

**Questions For Authors:**

Q1. I have a question regarding Theorems 4.9 and 4.12. Unlike Theorems 4.4, 4.6, 4.8, and 4.11, these two theorems do not include $p$ in their convergence rates. Why is this the case? Are these convergence rates tight? It seems more natural for these theorems to also depend on $p$.

Q2. In Theorem 4.9, is the step size choice $\eta_t = \frac{6 \log T}{\mu n T}$ correct? Compared to the step size in Theorem 4.8, this choice is smaller by a factor of $n$. Could you clarify this discrepancy?

**Relation To Broader Scientific Literature:**

While most prior work on shuffled SGD, such as results on random reshuffling, assumes Lipschitz-smooth gradients to establish convergence rates, this paper generalizes the framework by considering a weaker smoothness assumption. This generalization enhances the applicability of shuffling SGD, particularly in the context of neural network training.

**Theoretical Claims:**

I briefly checked the proofs of Theorems 4.4, 4.6, 4.8, and 4.9, and did not identify any significant flaws.

---

> ### Author Rebuttal · Authors · 2025-03-31
>
> Thank you very much for your detailed and helpful comments. Please find our responses below:
>
> **W1:** We appreciate your perspective but respectfully disagree that the paper is merely combining existing results. As we emphasize in Section 4.2, the main technical challenge arises specifically from analyzing the case when $t<\tau$. Each step in our analysis is conditioned not only on the event $t<\tau$ but also on all previous steps within the same epoch, due to the shuffling method. This dual conditioning has not been addressed by prior work that focuses only on either shuffling or $\ell$-smoothness.  Additionally, our results hold under the relaxed gradient assumption (Assumption 4.3), and even without it, thereby broadening their scope compared to existing $\ell$-smooth literature. We hope the reviewer can reconsider the novelty of our contribution in this context. Moreover, even if the novelty aspect were debated, considering the popularity of shuffling algorithms and restrictive nature of Lipschitz smoothness assumptions, our contributions extend theoretical guarantees to a broader class of problems. Thus, we respectfully suggest that novelty alone should not be considered a decisive reason for rejection.
>
> **W2:** We agree and will explicitly add the additional parameters for completeness. Thank you for this suggestion.
>
> **Q1:** Thanks for bringing up this important point. We apologize for any confusion caused. Indeed, complexities in Theorems 4.9 and 4.12 should not be directly compared to those in Theorems 4.8 and 4.11 due to the potentially large constant $G'$, which implicitly depends on $p$, as discussed in Section 4.3. The results in 4.9 and 4.12 are meant to be initial analyses for scenarios without the variance assumption. For completeness, if we did apply variance Assumption 4.3 and follow proof in Theorem 4.6, we could derive corresponding complexities of $\mathcal{O}(n^{\frac{p}{2}+1}\epsilon^{-\frac{1}{2}})$ and $\mathcal{O}(n^{\frac{p}{2}+1}\epsilon^{-\frac{3}{2}})$ for strongly convex and non-strongly convex cases respectively, with arbitrary shuffling scheme.
>
> **Q2:** Thank you for catching this typo. You are correct—the proper step size should be $\eta_t=\frac{6\log(T)}{\mu T}$. The learning rate currently listed in the draft mistakenly corresponds to the inner-loop step size. We appreciate your attention to details.

---

### Official Review · Reviewer_L7uE · 2025-03-14

**Overall Recommendation:** 3

**Summary:**

This paper studies convergence upper bounds of shuffling-based SGD on finite-sum minimization problems, focusing on random reshuffling SGD as well as SGD with arbitrary permutation (i.e., theorems that hold for any choice of permutations). The key contribution of this paper is the extension of standard (Lipschitz) smoothness assumption on the component functions to a generalized assumption named $\ell$-smoothness. For $\ell$-smooth functions with sub-quadratic $\ell$, the paper carries out convergence analysis of random reshuffling and arbitrary permutation-based SGD on nonconvex, strongly convex, general convex functions. Numerical experiments are carried out to compare performance of with-replacement SGD and shuffling-based SGD variants.

## Update after rebuttal
As I stated in my Rebuttal Comment, the authors' response addressed most of the questions I had, and I raised my score to 3 to reflect this. Having said that, I believe the paper has room for improvement in terms of clarity; I hope that the authors reflect the clarifications (e.g., on the dependence of different rates on $p$) in the next revision.

**Claims And Evidence:**

- Comments on theoretical claims are deferred to the "Strengths and Weaknesses" section.

- The authors present several experiments, but unfortunately the performance gap between with-replacement SGD vs shuffling-based variants does not look significant except Figure 1. I doubt it's fair to claim better performance of random-reshuffling SGD or fixed-shuffling SGD based on Figures 2 and 3. Also, for these experimental results to corroborate the theoretical results, random-reshuffling SGD should have converged the fastest across different settings; however, it seems that there is no clear winner.

**Essential References Not Discussed:**

The paper seems to include most of the essential references, but the authors should mention Mishchenko et al (2020) and Ahn et al (2020) when they discuss prior works on nonconvex optimization. Mishchenko et al are one of the first groups of people to prove convergence rates for nonconvex smooth optimization, and Ahn et al (2020) study nonconvex Polyak-Łojasiewicz functions. Also, it'd be more complete if the paper cites "Convergence of Random Reshuffling Under The Kurdyka-Łojasiewicz Inequality" by Li et al (2023).

**Experimental Designs Or Analyses:**

The design of experiments comparing the performance of with-replacement SGD and three variants of shuffling-based SGD looks quite standard to me, hence no issue identified.

**Methods And Evaluation Criteria:**

This paper does not propose a new method, and it analyzes existing methods theoretically under a relaxed set of assumptions. There are empirical evaluation results comparing the convergence speed of different methods, and I think the criteria for evaluation are sound.

**Other Comments Or Suggestions:**

- In Line 102, the term RGA is used before being introduced a few lines below.

**Other Strengths And Weaknesses:**

Strengths
- Extending the existing analyses of shuffling-based SGD to a wider function class of $\ell$-smooth functions is definitely meaningful.

- The authors honestly and directly discuss limitations of their results, which I appreciate.

Weaknesses
- First of all, to be strict, the paper violates the templates because the authors omitted the placeholder for author names. Unlike other papers, I don't see "Anonymous Authors" right below the title, as well as the footnote (attached to the placeholder) at the bottom of page 1.

- The biggest weakness of the results presented in this paper is the dependence of the epoch count $T$ on $1/\delta$. Each of the theorems on random reshuffling comes with an unfortunate $T = \Omega(poly(1/\delta))$ requirement on $T$, implying that for tiny choices of $\delta$ we would need to run a ton of epochs to meet the requirement. Given that dependences on $1/\delta$ in high-probability guarantees are typically poly-logarithmic, this is a big shortcoming in my opinion. The paper seems to build upon Theorem 5.3 of Li et al (2023a), and I understand that the existing theorem shares the same limitation. However, this doesn't mean that the improvement of poly to polylog dependence on $1/\delta$ is impossible, so the authors should discuss this in the paper.

- Some of the theorems have missing assumptions, although minor. In Theorem 4.11, we need to additionally assume that a finite $w_*$ exists. For Theorem 4.12, is $G'$ always guaranteed to be finite even when the sublevel set is unbounded? I guess so, but I'm not 100% sure.

- I find the proof sketch confusing. The authors claim that they demonstrate that smoothness is maintained with high probability along the training trajectory, but smoothness is a global property of a function, and has nothing to do with a specific trajectory.

**Questions For Authors:**

1. Is there any hope for improving dependencies on $1/\delta$?

2. For Theorem 4.9 and 4.12, why do these theorems have no dependence on $p$? Due to the absence of $p$, it looks to me that the epoch complexity of any arbitrary scheme is better than random reshuffling whenever $p > 1$, which does not sound right to me. Can you clarify why?

3. After most theorems, the authors discuss step size choices and the resulting epoch/gradient computation complexities. It is quite difficult to follow why these particular choices of step sizes are used and how different dependencies are derived. Specifically, I failed to derive the $-\frac{p}{2-p}$ terms that arise in the exponents of $\delta$ when specifying the dependency of $T$ on $1/\delta$. Can you elaborate on such derivations?

4. When talking about "one possible step size" in Lines 207 and 242, I see no dependence on $p$, which is different from the choice of $\eta$ in Lines 198 and 238 that involve $p$. Are the authors talking about the special case of $p=0$ here?

**Relation To Broader Scientific Literature:**

This paper studies popular variants of SGD, so it may have some broader impact on other scientific areas that involve optimization.

**Theoretical Claims:**

I unfortunately did not have the time to check the details of the proofs in the supplementary material. I hope the authors provide a better intuition on why the requirement on the number of epochs $T = \Omega (poly(1/\delta))$ is difficult to remove (which I point out below in the Strengths and Weaknesses section).

---

> ### Author Rebuttal · Authors · 2025-03-31
>
> Thanks for the careful reading and constructive feedback. Below we respond to each point in detail:
>
> **Q1:** Achieving a logarithmic dependency on $1/\delta$ remains an open challenge under $\ell$-smoothness, and establishing such a result would constitute a significant advancement. As noted in Remark 4.5, most existing works under $\ell$-smooth assumptions exhibit polynomial dependencies on $1/\delta$. The only exception is in Theorem 4.1 of [1], where independence between steps allows for a log dependence; however, they turn to polynomial dependence in Theorem 6.2 a few pages later, after the independence is lost. That means no log complexity has been achieved with dependence between steps and $\ell$-smoothness. Intuitively, under $\ell$-smoothness, the parameter $\delta$ accounts for bad cases from both variance and smoothness, whereas $L$-smooth scenarios only considers the former. Thus, significantly improving this dependency is nontrivial and would require substantial theoretical breakthroughs. We respectfully ask the reviewer to reconsider whether this aspect should be regarded as a major weakness, given the current state-of-the-art under similar assumptions.
>
> **Q2:** Thanks for highlighting this, we apologize for the confusion. The complexity bounds in Theorems 4.9 and 4.12 should not be directly compared to those in Theorems 4.8 and 4.11. The constant $G'$ in Theorems 4.9 and 4.12 can potentially be very large (as we mention in Section 4.3) and is implicitly influenced by $p$ through $G'$. These results serve as initial steps toward analyzing cases with no variance assumptions whatsoever. For completeness, if we were to adopt the variance Assumption 4.3 (similar to Theorem 4.6), we could derive results following proof of Theorems 4.6, yielding total gradient evaluation complexities of $\mathcal{O}(n^{\frac{p}{2}+1}\epsilon^{-\frac{1}{2}})$ and $\mathcal{O}(n^{\frac{p}{2}+1}\epsilon^{-\frac{3}{2}})$ for strongly convex and non-strongly convex cases with arbitrary scheme, respectively.
>
> **Q3:** To clarify the dependency of $T$ on $\delta$: note first that we have $H=\mathcal{O}(\delta^{-1})$, which implies $G=\mathcal{O}(\delta^{-\frac{1}{2-p}})$ and consequently $L=\mathcal{O}(\delta^{-\frac{p}{2-p}})$. Combining these with the constraints $\eta^3 T \leq \mathcal{O}(L^{-2})$ and $T \geq \frac{32\Delta_1}{\eta\delta\epsilon^2}$ gives the specified polynomial dependencies. Sorry for the confusion here, we will clarify this reasoning further in the appendix of a revision.
>
> **Q4:** Not really. The parameter $p$ is implicitly included within $L$ as we have $L=\mathcal{O}(n^{\frac{p}{2}})$.
>
> **W3:** Thanks for catching the missing assumption here, and $G'$ is finite indeed, but we need to add a short proof for that. We will add these in revision.
>
> **W4:** Sorry for the confusion from the wording. By "smooth along the training trajectory," we meant that smoothness conditions hold between consecutive points during training, not necessarily globally across the entire trajectory. We recognize this wording was imprecise and will clarify this point explicitly to avoid misunderstandings.
>
> We greatly appreciate the reviewer’s insights and suggestions, which have significantly improved the clarity of our manuscript.
>
> [1] Li, H., Rakhlin, A., Jadbabaie, A. (2023). Convergence of adam under relaxed assumptions.

---

> > ### Comment · Reviewer_L7uE · 2025-04-03
> >
> > Thank you very much for the response. The response addresses most of the questions I had, and I hope that the authors reflect the clarifications in the next revision (i.e., making the $p$ dependencies more explicit and making the RR vs arbitrary scheme rates more comparable). Although I still have some reservations due to the dependence on $1/\delta$, I feel more positive about the paper. I have decided to raise my score to 3.
> >
> > One more suggestion: The paper's title is a bit too general and does not clearly reflect the key messages/contributions. I recommend changing the title in the next revision.

---

> > > ### Author Response · Authors · 2025-04-08
> > >
> > > Thank you so much for the positive reconsideration and helpful suggestions! We will definitely address these points explicitly in the revised version and get a clearer title for the paper. Thanks again for your time and insights.

---

### Decision · Program_Chairs · 2025-05-01

**Decision:**

Accept (poster)

**Comment:**

This submission focuses on unconstrained minimization and aims to combine shuffling idea with generalized smoothness. The goal is to generalize the Lipschitzness assumption on the gradient that is often violated in applications.

This submission received 4 reviews and after the rebuttal period, all the reviewers agree on accepting the paper. The reviewers and myself agree that the problem is worth studying since both shuffling methods and generalized smoothness are common in applications and good theory on this front is interesting to the community.

On the other hand, there are also aspects of the paper that require improvements. For example, the polynomial dependence on the failure probability $\delta$ is noted by many of the reviewers (indeed it has been improved to a logarithmic dependence in the literature, without shuffling).

Moreover, the assumption that the authors have (Eq. 3) is stronger than the works that focus on shuffling without the generalized smoothness assumption. Indeed, it is possible for the variance assumption Eq. (3) to require bounded domains in many cases and then the appeal of generalized smoothness is significantly less. Moreover, the constant $G$ that is used in the paper seems to be much larger than the previous works.

On the presentation side, the writing needs to be polished and improved. There are not many explanations between different results and many inline equations make the text quite a bit hard to read. The subtleties such as the stronger variance assumption compared to the shuffling literature are not discussed in the text. Similarly, proper comparisons with the literature on generalized smoothness are also necessary, to compare the constant $G$, dependence on $p$, dependence on $\delta$, and so on.